# A new reproductive mode in anurans: Natural history of *Bokermannohyla astartea* (Anura: Hylidae) with the description of its tadpole and vocal repertoire

**Leo Ramos Malagoli**[1,2]*, **Tiago Leite Pezzuti**[3], **Davi Lee Bang**[4], **Julián Faivovich**[5,6], **Mariana Lúcio Lyra**[2], **João Gabriel Ribeiro Giovanelli**[2], **Paulo Christiano de Anchietta Garcia**[3], **Ricardo Jannini Sawaya**[7], **Célio Fernando Baptista Haddad**[2]

**1** Núcleo São Sebastião, Parque Estadual da Serra do Mar, Fundação para a Conservação e a Produção Florestal do Estado de São Paulo, São Sebastião, São Paulo, Brazil, **2** Departamento de Biodiversidade e Centro de Aquicultura (CAUNESP), Instituto de Biociências, Universidade Estadual Paulista, Rio Claro, São Paulo, Brazil, **3** Programa de Pós-Graduação em Zoologia, Instituto de Ciências Biológicas, Universidade Federal de Minas Gerais, Belo Horizonte, Minas Gerais, Brazil, **4** Programa de Pós-Graduação em Biologia Comparada, Departamento de Biologia/FFCLRP, Universidade de São Paulo, Ribeirão Preto, São Paulo, Brazil, **5** División Herpetología, Museo Argentino de Ciencias Naturales-CONICET, Buenos Aires, Argentina, **6** Departamento de Biodiversidad y Biología Experimental, Facultad de Ciencias Exactas y Naturales, Universidad de Buenos Aires, Buenos Aires, Argentina, **7** Centro de Ciências Naturais e Humanas, Universidade Federal do ABC (UFABC), São Bernardo do Campo, São Paulo, Brazil

\* lrmalagoli@gmail.com

**Data Availability Statement:** All relevant data are within the paper and its Supporting Information files.

## Abstract

Anurans have the greatest diversity of reproductive modes among tetrapod vertebrates, with at least 41 being currently recognized. We describe a new reproductive mode for anurans, as exhibited by the Paranapiacaba Treefrog, *Bokermannohyla astartea*, an endemic and poorly known species of the Brazilian Atlantic Forest belonging to the *B. circumdata* group. We also describe other aspects of its reproductive biology, that are relevant to understanding the new reproductive mode, such as courtship behavior, spawning, and tadpoles. Additionally, we redescribe its advertisement call and extend its vocal repertoire by describing three additional call types: courtship, amplectant, and presumed territorial. The new reproductive mode exhibited by *B. astartea* consists of: (1) deposition of aquatic eggs in leaf-tanks of terrestrial or epiphytic bromeliads located on or over the banks of temporary or permanent streams; (2) exotrophic tadpoles remain in the leaf-tanks during initial stages of development (until Gosner stage 26), after which they presumably jump or are transported to streams after heavy rains that flood their bromeliad tanks; and (3) tadpole development completes in streams. The tadpoles of *B. astartea* are similar to those of other species of the *B. circumdata* group, although with differences in the spiracle, eyes, and oral disc. The vocal repertoire of *B. astartea* exhibits previously unreported acoustic complexity for the genus. *Bokermannohyla astartea* is the only bromeligenous species known to date among the 187 known species within the tribe Cophomantini. We further discuss evolutionary hypotheses for the origin of this novel reproductive mode.

**Funding:** Funding: This work was part of L. R. M.'s PhD degree. L. R. M. thanks Conselho Nacional de Desenvolvimento Científico e Tecnológico (CNPq) for doctoral fellowship (grant #141259/2014-0). For financial support, we thank São Paulo Research Foundation (FAPESP) (grants #2008/54472-2, #2008/50928-1, #2013/50741-7, #2014/50342-8, and #2014/23677-9). T. L. P. thanks Coordenação de Aperfeiçoamento de Pessoal de Nível Superior (CAPES) for PROTAX fellowship (grant #440665/2015-9) and CNPq for "Ciência Sem Fronteiras" fellowship (grant #202081/2015-0). D. L. B. thanks FAPESP for doctoral fellowship (grant #2017/27137-7). J. F. thanks Agencia Nacional de Promoción Científica y Tecnológica (ANPCyT PICT 2015-820), and FAPESP (grant #2018/15425-0). M. L. L. thanks FAPESP for fellowship (grant #2017/26162-8). J. G. R. G. thanks CAPES for doctoral fellowship (Financing Code 001). C. F. B. H., R. J. S and P. C. A. G. thanks CNPq for research productivity fellowship (grants # 306623/2018-8, #312795/2018-1, #310301/2018-1, respectively). The funders had no role in study design, data collection and analysis, decision to publish, or preparation of the manuscript.

**Competing interests:** The authors have declared that no competing interests exist.

## Introduction

The transition from water to land was a major evolutionary event in vertebrate history and it imposed challenges to morphological and physiological mechanisms underlying fundamental biological processes such as reproduction [1, 2]. Terrestriality in tetrapods is mainly driven by selective pressures such as aquatic predation and vacant niches in terrestrial environments [3, 4]. However, recent studies indicate that sexual selection by male-male competition [5] and parental care [6], also played an important role in the evolution of terrestrial reproductive modes in anuran amphibians, indicating that multiple factors can contribute towards terrestriality. Despite retaining an ancestral dependence on water [7, 8], amphibians are peculiar by having the greatest diversity of reproductive modes among tetrapods [7], which is often paralleled by an array of behaviors [7, 8]. Anurans exhibit profound variation in reproductive modes, particularly in the tropics [7, 9], with at least 41 currently known modes globally [10–12]. Such diversity stems from complex life histories encompassing peculiarities in traits such as oviposition site, egg and clutch characteristics, rate and duration of larval development, stage and size of hatchlings, and type of parental care, if any [8, 13]. These reproductive modes range from fully aquatic to fully terrestrial and arboreal modes [9, 11, 14].

More specifically, arboreal reproductive modes include using phytotelmata as a site for reproduction [9, 11, 15], with the rainwater that accumulates in such plants providing benefits in the form of shelter and avoidance of predation and niche competition, which are found in other more open microhabitats [16]. However, low levels of oxygen, detritus accumulation and limited space may impose severe challenges for a species to thrive in a such microhabitat [7]. A widespread and common phytotelmata microhabitat used by frogs to reproduce is leaf-tanks of epiphytic or terrestrial bromeliads [15, 17].

Inherent to anuran reproduction is diversity in tadpole morphology and acoustic communication [7, 18]. Tadpoles in most cases have specialized morphologies depending on the reproductive mode and the habitat in which they develop [18]. Exotrophic tadpoles, for example, vary in body and tail shapes and oral disc configuration according to the environment in which they occur (*e.g.*, lotic and lentic water bodies). Differences in larval morphology may be considerable even in closely related species with distinct reproductive modes (*e.g.*, [19–21]). Likewise, acoustic communication represents an important evolutionary phenotype among major clades of terrestrial vertebrates, including anuran amphibians [22]. Anurans use vocal signals as the main form of communication in a myriad of social interactions [7, 22]. The main type of vocalization that is emitted by males is the advertisement call, which serves mainly to attract females for reproduction [7, 23]. However, other call types such as courtship, amplectant, and territorial may have unique traits and play determinant roles within the social context and reproductive biology of anurans [7, 23, 24]. The recognition of these call types improves our knowledge of behavior and sexual selection mechanisms (*e.g.*, [7, 25, 26]).

Primary data on reproductive biology are crucial for evolutionary studies of anurans (*e.g.*, [5, 15, 27, 28]), but they are still scanty for several Neotropical frog genera [7, 11]. This is the case for *Bokermannohyla* Faivovich, Haddad, Garcia, Frost, Campbell & Wheeler, 2005, a Brazilian endemic genus of gladiator treefrogs belonging to the tribe Cophomantini [29–32]. Thirty *Bokermannohyla* species are allocated among three species groups—the *B. circumdata* group, the *B. martinsi* group, and the *B. pseudopseudis* group—and are distributed throughout the Atlantic Forest, Cerrado, and Caatinga [31, 33, 34] morphoclimatic domains (*sensu* Ab' Saber [35]). Their reproduction is mostly associated with lentic and/or lotic environments such as ponds and streams, respectively [9, 34]. To date, detailed information on reproductive biology and natural history are available for only four species of the genus [36–39]. On the

other hand, tadpoles and advertisement calls are well known for most species of *Bokermanno-hyla* [40–42], with only a few species remaining for which these traits are unknown.

The Paranapiacaba Treefrog, *Bokermannohyla astartea* (Bokermann, 1967), belongs to the *B. circumdata* group [29] and is distributed throughout the Atlantic Forest morphoclimatic domain in the Serra do Mar, a long system of mountain ranges and escarpments in Southeast Brazil [31]. It is known to use epiphytic or terrestrial bromeliads located close to temporary pools and streams as vocalization sites [43–45]. Although *B. astartea* was described more than 50 years ago, detailed information on its biology and tadpole have remained unknown except for the brief mention of oviposition and tadpole development in bromeliads by Verdade *et al.* [46], and a brief description of the species advertisement call by Heyer *et al.* [44]. Over the course of 11 years, we obtained data on the natural history of *B. astartea* and found that it has a previously undescribed reproductive mode [7, 9, 11], which we describe herein. Furthermore, we describe other aspects of the reproduction of *B. astartea* that are strongly related to the new reproductive mode, including courtship behavior, spawning, and tadpoles. Lastly, we redescribe the species' advertisement call and extend its vocal repertoire by describing three new call types: courtship, amplectant, and presumed territorial. Our results reveal unique and novel characteristics for the reproductive biology of *Bokermannohyla* and the tribe Copho-mantini, and contribute to the knowledge of the diversity and evolution of reproductive modes in frogs.

## Materials and methods

### Study area

The study was conducted in the Núcleo Curucutu (NC) of Parque Estadual da Serra do Mar (PESM) (23˚59'8.29"S, 46˚44'37.11"W), a protected area similar to those in Category II or the International Union for Conservation of Nature (*i.e.*, [47]). The NC covers parts of the munici-palities of São Paulo, Itanhaém, Mongaguá, Juquitiba, and São Vicente, in the state of São Paulo, Southeast Brazil [48]. The area encompasses a mosaic of Atlantic Forest phytophysiog-nomies, such as highland grasslands and high montane forests [49, 50]. The NC is located about 50 km straight-line-distance from Reserva Biológica do Alto da Serra de Paranapiacaba [48], the type locality of *Bokermannohyla astartea* [43].

The study was carried out in strict accordance with the recommendations in the Guide for the Care and Use of Laboratory Animals of the National Institutes of Health. Field procedures and methods were approved by the Ethics Committee on Animal Use of the Universidade Estadual Paulista, Rio Claro, São Paulo, Brazil (CEUA/UNESP) (#036/2015 and #017/2016). Permits to work and collect in NC of PESM were granted by Comissão Técnico-Científica do Instituto Florestal (COTEC/IF) (processes #40.452/2004, #40.574/2006, and #260108–003.523/2014). Permits to capture and collect live specimens were granted by Instituto Chico Mendes de Conservação da Biodiversidade (ICMBio) (license numbers #019/07, #16350–1, #45665–1, #45665–2, and #45665–3).

### Data collection in the field

**Breeding and habitat use data.**    Field sampling in the NC, which usually occurred during the reproductive season (September to March), took place from 2005 to 2016 and totaled 71 days and 356 hours of field work exclusively to observe *B. astartea*. Samplings began at 1800 h and finished at 2400 h. Observations occurred at two breeding sites (BS) separated by 3.2 km straight-line-distance: BS1 (23˚59'34.35"S, 46˚42'55.34"W, municipality of São Paulo, 800 m above sea level [a.s.l.]), and BS2 (23˚59'46.00"S, 46˚44'45.87"W, municipality of Itanhaém, 820 m a.s.l.; Datum WGS84 for both sites) (Figs 1 and 2). Six sets of bromeliads were selected at

BS1: four sets of ground bromeliads on the margin of a temporary stream and two sets of epiphytic bromeliads above the margin of a permanent stream. The sets of bromeliads at BS1 were separated by a distance of 5–60 m. Six sets of bromeliads were also selected at BS2: four sets of epiphytic bromeliads above the margin of a permanent stream and two sets of epiphytic bromeliads above the margin of a temporary stream. The sets of bromeliads at BS2 were separated by 6–80 m. A tape-measure (in meters) was used to measure the horizontal distance of each set of bromeliads from water and the height of each set of bromeliads above water, as well as the width of the streams.

The sex of individuals was determined by the presence of developed prepollex, vocal sac and vocal slits, and vocal activity in males, and by visualization of oocytes in the inguinal region of females, by transparency of the belly and flanks, as well as the absence of a prepollex. Some adult exemplars were collected and euthanized via anesthetic overdose of 5% lidocaine. Liver and muscle samples were removed immediately thereafter and kept in 100% ethanol for DNA extraction protocols. These samples were also used for tadpole molecular identification (see details below). Individuals were fixed in 10% formalin and preserved in 70% ethanol as vouchers. These specimens were measured with digital calipers (0.05 mm precision) in order to determine if mean snout-vent lengths (SVL) differed between resident and satellite males (see below), as well as between males and females, which were evaluated by two-tailed Student's t-tests with a significance level of $\alpha = 0.05$ [51] in the R platform [52]. Additional data on adult SVL were obtained from specimens deposited in the following collections: Coleção de Anfíbios "Célio F. B. Haddad", Departamento de Biodiversidade, Instituto de Biociências, Universidade Estadual Paulista (CFBH); Museu de Zoologia da Universidade de São Paulo (MZUSP); Museu Nacional da Universidade Federal do Rio de Janeiro (MNRJ); Coleção de Anfíbios, Museu de Zoologia, Universidade Estadual de Campinas (ZUEC-AMP); and Coleção Herpetológica da Universidade Federal de Minas Gerais (UFMG) (S1 Appendix).

**Behavioral data.** Focal animal and all-occurrence sampling methods were used for observation of adult behaviors [53, 54], which record all types of behaviors performed by a species during a given observation period. All observations were made with a head lamp with a red filter to minimize the influence of researcher presence during samplings (*e.g.*, [37, 55]). Leica D-Lux 4® and Canon PowerShot G1X Mark II® digital cameras were used to record and film behavioral repertories. Some males exhibit patterns of spots and blotches on the dorsum [43, 44], which made possible their individualization (S1 Fig) during sampling from September 2014 to March 2015. Each individualized male received a number and had its dorsum photographed. A field photographic guide of all photographed exemplars was produced for the identification of individuals. Males were individualized in order to obtain information on site fidelity (*e.g.*, [56–59]). Observations were also made about male territoriality and satellite behavior (*sensu* Wells [60, 61]). Distances between some resident males (calling males) and satellite males (non-calling males) were measured with a measuring tape (in centimeters).

**Spawn, egg, and tadpole data.** The search for spawns, eggs and tadpoles were made both in bromeliads and in streams. Spawns found in bromeliads were counted and some eggs were measured to the nearest 0.05 mm with an analog caliper. Tadpoles were counted and observed in bromeliads and also in temporary streams and backwaters of permanent streams, always below or near the sampled bromeliads. Spawns and tadpoles in bromeliads were captured/collected with a teaspoon, while tadpoles in streams were captured with a dip-net. Some samples of spawns and tadpoles were collected and euthanized by the same method described above but were kept preserved in 10% formalin. Preserved eggs and oocytes were measured with an ocular micrometer in a Zeiss stereomicroscope. Some tissue samples obtained from tadpole tails and entire eggs were kept in 100% ethanol for DNA sampling.

Voucher specimens of adults, tadpoles and spawn, as well tissue samples are housed in CFBH collection (S1 Appendix).

**Vocal repertoire data.**   Recordings of the vocal repertoire, including advertisement, courtship, amplectant, and presumed territorial calls, were obtained with an external unidirectional Sennheiser ME-66 microphone coupled to a Marantz PMD-660 digital audio recorder adjusted to a sampling rate of 48 kHz and 16-bit resolution. All recordings were obtained with direct visualization of calling individuals and air temperature was measured with a digital thermometer. Audio recordings are housed in Fonoteca Neotropical Jacques Vielliard (FNJV), Museu de Zoologia, Instituto de Biologia, Universidade Estadual de Campinas (S1 Table).

## Molecular identification of tadpoles

Molecular data were used to confirm the identity of *B. astartea* tadpoles found in both bromeliads and in temporary and permanent streams. This procedure was conducted to differentiate *B. astartea* from two other sympatric *Bokermannohyla* species whose tadpoles also occur in NC streams, *B. circumdata* and *B. hylax* [48], and to support subsequent morphological analysis. Total DNA was extracted using a standard ammonium acetate precipitation method [62]. A fragment of the mitochondrial 16S ribosomal gene (16S) was amplified using primers 16Sar-L and 16Sbr-H [63] and the PCR cycling conditions described in Lyra *et al.* [64]. PCR products were purified using enzymatic reaction and sequenced by Macrogen Inc., Republic of Korea. DNA sequences were quality-verified and trimmed using Geneiuos V.6 [65]. A local database was built for tadpole identification, which included all 16S sequences available for Anura in GenBank along with newly generated sequences for adult *B. astartea* from three different localities in the state of São Paulo (accession numbers: MH20128–MH201230; Núcleo Curucutu, Tapiraí, and São Miguel Arcanjo). The tool blastn (BLAST+ application; [66]) was then used to compare the tadpole sequences with the database, considering a value of 1.0e-13. Values of 98–100% similarity were considered sufficient to assign a tadpole to a species. Tadpole identification, along with metadata including voucher, GenBank accession numbers, and collection breeding site, are given in S2 Table.

## Procedures for tadpole description

A total of 29 tadpole specimens from BS1 in developmental stages between 25 and 37 of Gosner [67] (lots CFBH 38055, 42649, 42650, 42651, 42652, 42653, 42654, 42655, and 42659; details of lots are provided in S1 Appendix) were analyzed for qualitative external morphological characterization. Some analyzed lots had representatives confirmed by molecular identification (S2 Table). Eight tadpoles in stages 35–37 were used to describe external morphology, including measurements and proportions (lots CFBH 42649, 42650, 42651, and 42652). Additional lots not used in the description were used to describe intraspecific variation. Terminology and measurements for external morphology follow Altig & McDiarmid [19] for total length (TL), body length (BL), tail length (TAL), maximum tail height (MTH), internarial distance (IND), interorbital distance (IOD), tail muscle width (TMW), and tail muscle height (TMH); Lavilla & Scrocchi [68] for body width (BW), body width at narial level (BWN), body width at eye level (BWE), body height (BH), eye-snout distance (ESD), eye-nostril distance (END), nostril-snout distance (NSD), eye diameter (ED), narial diameter (ND), snout-spiracular distance (SSD), and oral disc width (ODW); Grosjean [69] for dorsal fin height (DFH) and ventral fin height (VFH); Lins *et al.* [40] for spiracle length (SL), spiracle distal edge height (SDH), dorsal fin insertion angle (DFiA), oral disc position (ODP), and length of anterior gap in marginal papillae of the oral disc (AGL). Vent tube length was also measured, as the distance between the anterior insertion of the ventral wall of the tube in the body and its most distal margin, taken in ventral view (VTL). Tadpoles were photographed immersed in water using

an adjustable platform to support the specimens in lateral, dorsal, and ventral views [70]. Lateral line system descriptions and corresponding terminology follow Lannoo [71]. All measurements were taken to the nearest 0.1 mm with the aid of ImageJ version 1.50b [72].

### Acoustic analyses

A total of 23 audio recordings of 15 males in uncompressed mono.wav formats were used to characterize the vocal repertoire of *B. astartea*. Analyses were conducted in Raven Pro v1.5 [73], using the following spectrogram parameters: window type Hann and size of 256 samples, which resulted in a 3 dB filter bandwidth of 270 Hz, overlap at 90% (locked; hop size = 0.54 milliseconds [ms]) for time grid, and DFT of 1024 samples (grid spacing = 46.9 Hz) for frequency grid. Recordings were filtered at up to 200 Hz to reduce background noise. All other settings followed software default values. Measurements were obtained through both oscillograms for temporal traits and spectrograms for spectral traits. Figures were produced using seewave 1.7.6 [74] and tune R 1.3.2 [75] packages in the R platform [52], using the following parameters: window type = Hann, overlap = 90%, FFT size = 256 points. The following acoustic traits were measured according to Köhler *et al.* [23], except for call rate, which was adapted from Cocroft & Ryan [76]: (1) call duration or note series duration as time elapsed from the beginning until the end of a call or a note series (for advertisement calls); (2) number of notes per call or series; (3) call rate per minute, as the total number of calls in a series -1 /duration from the beginning of the first note to the beginning of the last note; (4) note duration, as time from the beginning to the end of a single note within a call; (5) number of pulses per note; (6) pulse rate, as the quotient between number of pulses and note duration; (7) note rise time as the percentage at which a note reaches its maximum amplitude, obtained with the "Peak Time" function in Raven Pro v1.5 [73], divided by note duration and multiplied by 100; (8) interval between calls or notes, as time between the end of one call or note to the beginning of the subsequent call or note; (9) dominant frequency as the peak of energy within a note obtained with the "Peak Frequency" parameter in Raven Pro v1.5 [73]; (10) pulse frequency modulation measured for calls that were assigned as having a presumably territorial function (see Results), defined as the change in frequency over time and obtained as dominant frequency of the last portion minus that of the first portion of each pulse.

Descriptive statistics are henceforth shown as mean ± standard deviation (minimum–maximum [when applicable], and number of individuals or specific records).

## Results

### Temporal breeding pattern and habitat use

*Bokermannohyla astartea* is a prolonged breeder (*sensu* Wells [58]). Calling males were observed from September to March during the rainy season. Adult females were observed from October to March. No males or females were observed from April to August in the dry season. Males initiate vocalization activities after sunset between 1800 and 1900 h and decrease activity after 2400 h. Among males observed in calling activity (n = 89), most (93.3%, n = 83) used ground and epiphytic bromeliads at heights ranging 0.20–2.4 m above the water as calling sites (Figs 1 and 2). The bromeliads used belong to the genera *Nidularium* (n = 43), *Vriesea* (n = 34), and the species *Wittrockia cyathiformis* (n = 6). Males vocalizing outside bromeliads (6.7%, n = 6) used shrubs close to bromeliads. Males were observed using different portions of bromeliads as calling sites. Most used lateral leaf-tanks (57.3%, n = 51), but some used central tanks (23.5%, n = 21), while others were observed positioned on bromeliad leaves outside of leaf-tanks (12.3%, n = 11), although they moved

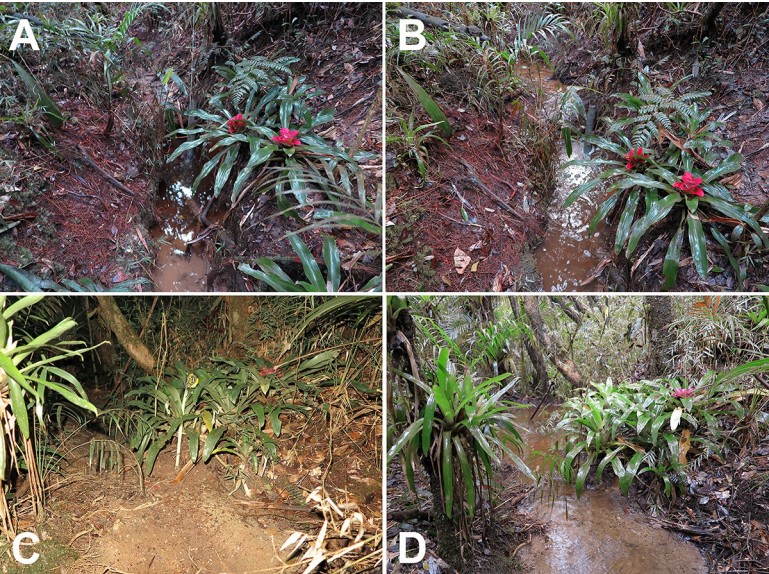

**Fig 1. Aspects of breeding site 1.** (A) and (C) temporary streams without rain; (B) and (D) the same sites after rain.

among spots during the night. Horizontal distance between the studied bromeliads and stream water ranged 0.20–1 m (n = 12). Height of bromeliads above stream water varied 0.20–3 m (n = 12). Stream width varied 0.5–5 m (n = 8).

The SVL of adult males measured 39.7 ± 1.3 mm (36.8–42.7 mm, n = 40) while that of adult females measured 43.3 ± 0.8 mm (42.3–44.3 mm, n = 4). Males and females differed in SVL, with females being larger than males (t = 5.34, df = 41, P < 0.001).

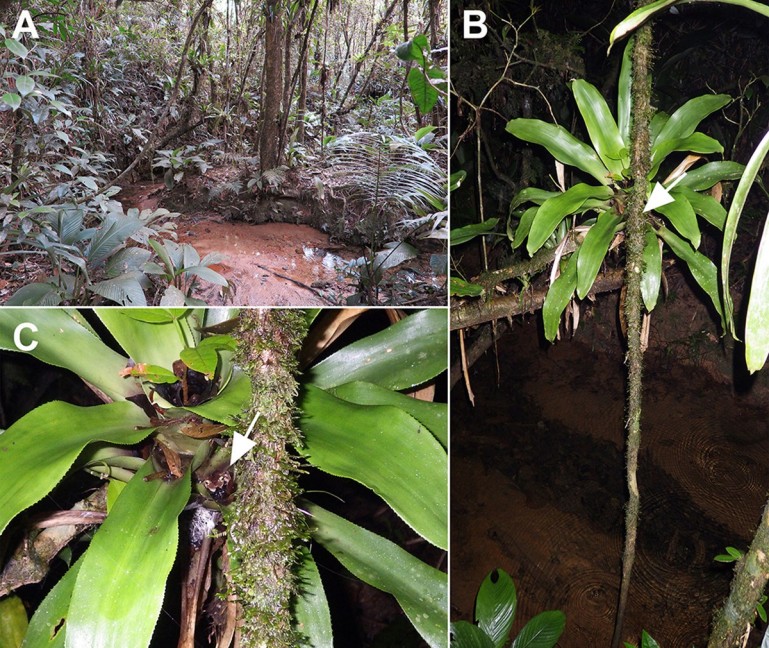

**Fig 2. Aspects of breeding site 2.** (A) A permanent stream inside the forest. (B) Epiphytic bromeliad over a stream. The white arrow indicates the exact location of a male of *B. astartea* in calling activity. (C) Detail of the male of *B. astartea* in a bromeliad leaf-tank. The white arrow indicates the position of a male of *B. astartea* in calling activity.

## Male site fidelity, satellite behavior, and territoriality

A total of 11 males were individualized according to their dorsal patterns of blotches and spots (S1 Fig). Seven were individualized at BS1 and four at BS2. Eight individualized males were recaptured in their respective set of bromeliads; five at BS1 and three at BS2. The interval between recaptures ranged 1–53 days (Table 1). Satellite males (non-calling males) were found close to resident males (calling males), with the distance between them being 86 ± 42 cm (20–150 cm, n = 22). Unlike resident males that generally vocalized in leaf and central tanks of the bromeliads, satellite males remained on bromeliad leaves or in shrubs near the bromeliad, and adopted a crouching posture (Fig 3A). There was no significant difference in SVL between resident males (39.7 ± 1.0 mm, 38.5–41.4 mm, n = 6) and satellite males (41 ± 1.3 mm, 39.1–41.97 mm, n = 4) (t = 1.183, df = 8, P = 0.103). Two males previously exhibiting satellite behavior were subsequently observed assuming a resident posture by starting to vocalize in leaf-tanks. Some males exhibited scratch scars on the dorsum and head (n = 14, 15.7% of all males observed in the study), as in individualized male number 1 (S2 Fig).

## Courtship and oviposition behaviors

Five courtship sequences were observed, two until oviposition. All these courtships were observed at BS1, but in different sets of bromeliads. Courtships 1, 2, 3, and 5 were recorded in sets of ground bromeliads located on the banks of a temporary stream separated by distances of 5–15 m. Courtship 4 was recorded in a set of epiphytic bromeliads about 1.2 m above the water at the margin of a permanent stream. The courtship behavior of *Bokermannohyla astartea* is composed of a sequence of steps as detailed below (see S3 Table for complementary information).

**Courtship 1.** A resident male was vocalizing in a leaf-tank when a female approached and observed the male for about 3 minutes. The female sequentially inspected four leaf-tanks, each of them for 2–4 minutes. The female then entered in the same leaf-tank where the male was vocalizing and remained there for about a minute without amplexus. After this period, the female left the leaf-tank jumping toward a nearby shrub, not returning to the bromeliad.

**Courtship 2.** A female (CFBH 38045) approached a resident male vocalizing in a bromeliad leaf-tank. A satellite male was located about 60 cm away (Fig 3A). When the female started to enter the leaf-tank of the vocalizing resident male, the satellite male jumped towards the pair. The female fled, jumping several times towards another set of bromeliads 6 m away. The resident male, apparently by mistake, amplected the satellite male. The satellite male was released after about two minutes.

**Courtship 3.** The same female from courtship 2 inspected a leaf-tank in another bromeliad while a male (CFBH 38044) in a nearby leaf-tank was emitting courtship calls. The female

**Table 1. Males of *Bokermannohyla astartea* recaptured in the same sets of bromeliads in the two breeding sites sampled (BS1 and BS2).**

| Individual's number | Number of recaptures | Days between recaptures | Breeding site (BS) |
|---|---|---|---|
| 1 | 4 | 9.2 (1–18) | BS1 |
| 2 | 4 | 9.2 (1–18) | BS1 |
| 3 | 5 | 16.8 (1–53) | BS1 |
| 6 | 1 | 16 | BS2 |
| 7 | 2 | 31 (16–46) | BS2 |
| 8 | 1 | 52 | BS1 |
| 10 | 3 | 27 (19–33) | BS2 |
| 11 | 3 | 9.3 (1–26) | BS1 |

Days between recaptures are shown as mean (range).

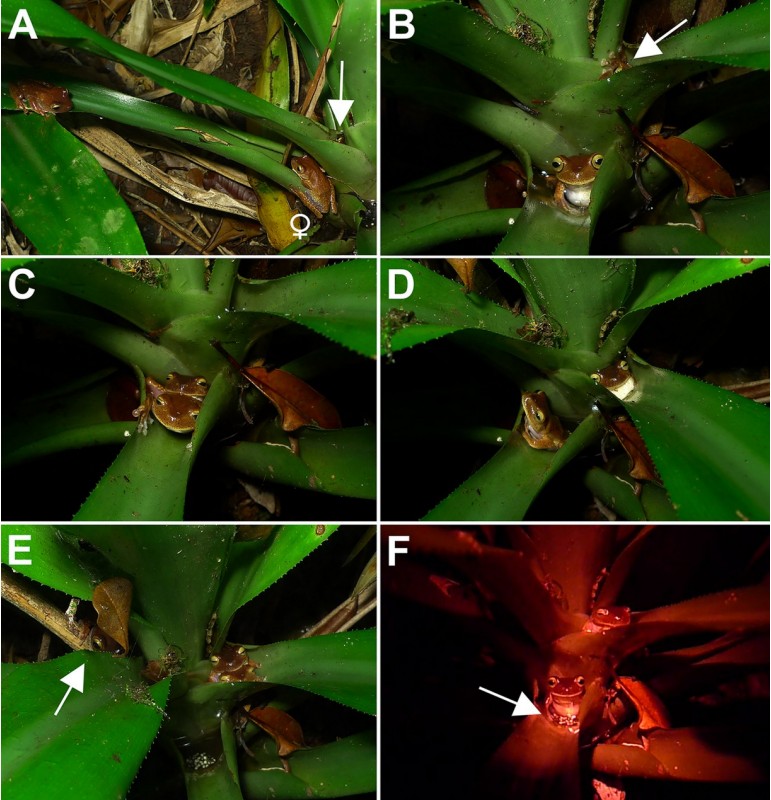

**Fig 3. Reproductive behaviors of *Bokermannohyla astartea*.** (A) Female approaches the resident male in vocal activity. The white arrow indicates the location of the resident male in the bromeliad leaf-tank. The satellite male was on the left observing the interaction. (B) In another set of bromeliads, the same female was inspecting the bromeliad leaf-tank while the resident male was in the leaf-tank above (white arrow). (C) Pair in amplexus in the leaf-tank inspected by the female. (D) After egg laying was completed, the resident male left the female and sat in the upper leaf-tank, where he began to emit a courtship call to attract the female again. (E) Pair in amplexus in the upper leaf-tank. Notice the spawn just placed in the anterior leaf-tank and the approach of the satellite male (white arrow). (F) Satellite male sitting on the newly deposited spawn (white arrow). The pair remains in amplexus in the leaf-tank above.

then entered the leaf-tank of the vocalizing male and axillary amplexus occurred. After two minutes, the female separated from the male and entered a different leaf-tank just below. The female inspected the bromeliad leaf-tank by sitting in the water for 5 minutes, while the male continued to emit courtship calls in the upper leaf tank (Fig 3B). The male entered the leaf-tank where the female was and they performed axillary amplexus again for about eight minutes. During amplexus and egg deposition, the male squeezed the female with his arms (Fig 3C and S1 Movie). Eighty-two eggs were laid during this episode of amplexus. The male then left the female and entered the upper leaf-tank, where he resumed vocalizing (Fig 3D). After three minutes, the female entered the leaf-tank where the male was calling and the pair resumed amplexus. At this moment, a satellite male (CFBH 38046) started approaching the pair, observing them closely (Fig 3E). Then, while the pair was in amplexus, the satellite male entered the leaf-tank of the first spawn and sat on it for about two minutes (Fig 3F and S2 Movie). The satellite male then left the leaf-tank of the spawn and approached the leaf-tank with the pair, where it watched the pair and passed under the leaf-tank, while the resident male emitted an amplexus call (S2 Movie). The last amplexus lasted about 12 minutes with 26 eggs laid. The female then left the resident male, who unsuccessfully still tried to hold her. The male returned

to the leaf-tank and continued to emit courtship calls. The satellite male remained perched in a leaf-tank just below (S3 Movie).

**Courtship 4.** This courtship was similar to courtship 3, but with the absence of satellite males. The female inspected three leaf-tanks before entering a fourth leaf-tank containing a male and the two performed amplexus. The female entered into amplexus twice with the same male in two different leaf-tanks of the same bromeliad. The first amplexus lasted 11 minutes and the female laid 57 eggs, while the second amplexus lasted 9 minutes and the female laid 49 eggs.

**Courtship 5.** This courtship was similar to courtship 1, except that amplexus occurred. During amplexus, the male seemed to rub his gular region on the head of the female. After about 30 minutes, the female left the leaf-tank abandoning the male; oviposition did not occur.

An additional 11 ongoing-amplexus were observed, but without observing the beginning and the end. In these cases, amplexus was always in a lateral leaf-tank of a bromeliad.

## Spawn, eggs, and tadpoles in bromeliads and in streams

Newly deposited spawn consists of aggregated eggs (Fig 4A), which after approximately three hours (n = 2) became a gelatinous mass (Fig 4B). Spawns were always found in bromeliad leaf-

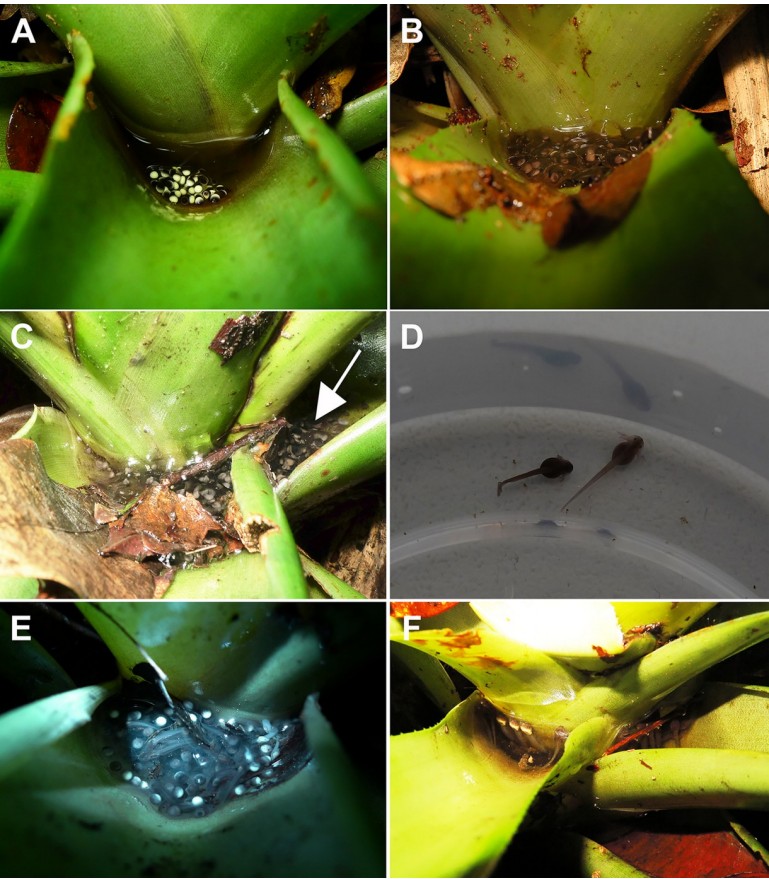

**Fig 4. Spawns and tadpoles of *Bokermannohyla astartea* in bromeliad leaf-tanks.** (A) Freshly laid spawn. (B) Gelatinous spawn a few hours after laying. (C) Spawn in different leaf-tanks of the same bromeliad. The white arrow indicates the spawn also in the lateral leaf-tank. (D) Newly hatched tadpoles with external gills. (E) Tadpoles amidst unfertilized eggs and other detritus. (F) Tadpoles in leaf-tanks full of water exhibiting the snout pointing upward near the water surface.

tanks, and never in the streams. Spawns had 45.3 ± 14.1 eggs (24–84 eggs, n = 64 spawns). Egg diameter including the gelatinous capsule was 8.8 ± 0.5 mm (7.8–9.8 mm, n = 145 eggs from 15 spawns) and without it 2.2 ± 0.2 mm (1.9–2.7 mm, n = 23 eggs, CFBH 42660). More than one leaf-tank containing spawn per bromeliad was often observed (Fig 4C). Three dissected females exhibited 172.3 ± 29.3 (139–194) mature oocytes. One of the females (CFBH 38045) deposited 108 eggs in two leaf-tanks of the same bromeliad (82 and 26 eggs, respectively). Considering mature oocytes and laid eggs, this female had 247 oocytes. Oocyte diameter was 1.85 ± 0.13 mm (1.58–2.06 mm, n = 58). The oocytes are black at the animal pole and beige at the vegetal pole. Spawns were found from October to March.

Newly hatched tadpoles were in stages 23–24 [67], had external gills (Fig 4C), and began feeding two or three days after hatching. Tadpoles were observed in leaf-tanks at stages 23–26 (n = 184 tadpoles, in 39 leaf-tanks of 17 bromeliads). The number of tadpoles in each leaf-tank was 4.71 ± 3.05 (1–16, n = 184 tadpoles, in 39 leaf-tanks of 17 bromeliads). The leaf-tanks contained detritus, such as leaves and seeds, as well as non-fertilized eggs undergoing decomposition, which the tadpoles bit and scraped, apparently feeding on them (Fig 4E). When leaf-tanks contained a lot of water the included tadpoles were quiescent with the snout pointing upward near the water surface (Fig 4F). Tadpoles were found in bromeliads from October to mid-April. Tadpoles at stages 26–36 (n = 23) were recorded in streams from January to March. The narrower streams (0.5–1 m of width) were temporary, with flow ceasing during the driest period, when remnant small pools served as environments for tadpole development. The broader streams (3–5 m of width) were usually permanent, and tadpoles were observed in backwaters amidst dead leaves and sediment. The substrate of the broader streams was generally composed of rocks and sand.

Samples of tadpoles collected from bromeliads were all identified as *B. astartea* based on DNA comparisons. On the other hand, in addition to *B. astartea*, tadpoles of syntopic species of the tribe Cophomantini were recorded in streams, namely *Aplastodiscus* aff. *albosignatus* and another species of *Bokermannohyla* (S2 Table). Tadpoles of *B. circumdata* and *B. hylax* could not be differentiated based on reference sequences available in GenBank, and 21 sequenced individuals were very similar molecularly (maximum pairwise distance 2%). Considering that differentiating between *B. circumdata* and *B. hylax* is out of the scope of this study, all these samples were considered *Bokermannohyla* gr. *circumdata* (S2 Table).

## Description of a new reproductive mode

The reproductive mode of *Bokermannohyla astartea* consists of: (1) deposition of aquatic eggs in water accumulated in leaf-tanks of ground or epiphytic bromeliads located on or over margins of temporary and/or permanent streams; (2) exotrophic tadpoles remain in the leaf-tanks during early stages (until stage 26), when they presumably jump or are transported to the streams after heavy rains that flood their bromeliad leaf-tanks (see Figs 1C, 1D, 2B and 2C); and (3) tadpole development completes in streams.

## Tadpole description

Ranges of measurements (in mm) and angles of tadpoles of *Bokermannohyla astartea* from BS1 are presented in Table 2.

**External morphology.** Body depressed (BH/BW = 0.78–0.85; Fig 5A–5D), BL 0.28–0.29 times TL; elongated elliptical in dorsal view, with slight lateral constrictions at eye level; in lateral view, contour flat in the peribranchial region and convex in the abdominal region. Snout rounded in dorsal (BWN/BWE = 0.72–0.83) and lateral views. Nostrils small (ND/BL = 0.034–0.042), elliptical, dorsally positioned (IND/BWN = 0.46–0.55), dorsolaterally directed, located

**Table 2. Ranges of measurements (in mm) and angles of tadpoles of *Bokermannohyla astartea* from breeding site 1, considering stages or ranges of stages of Gosner [67].**

| Lots | CFBH 42649 | CFBH 42650 | CFBH 42651 | CFBH 42652 |
|---|---|---|---|---|
| **Stages (n)** | **35–36 (2)** | **36 (2)** | **35–37 (2)** | **35–36 (2)** |
| **Measurements** | | | | |
| **Total length** | 50.7–51.5 | 51.5–51.6 | 50.7–52.2 | 52.6–52.8 |
| **Body length** | 14.7–14.8 | 14.0–14.0 | 14.7–14.9 | 14.7–15.4 |
| **Tail length** | 35.9–36.8 | 36.9–37.0 | 36.0–37.3 | 37.3–38.1 |
| **Maximum tail height** | 8.0–9.7 | 8.3–8.6 | 8.1–8.3 | 9.0–9.3 |
| **Dorsal fin height** | 2.9–3.3 | 2.9–3.2 | 3.0–3.0 | 3.3–3.5 |
| **Ventral fin height** | 2.3–2.6 | 2.2–2.4 | 2.0–2.2 | 2.3–2.5 |
| **Tail muscle height** | 5.0–5.1 | 5.1–5.2 | 4.9–5.2 | 5.3–5.5 |
| **Body height** | 7.6–7.8 | 7.6–7.9 | 7.1–7.6 | 7.5–8.6 |
| **Body width** | 1.5–1.6 | 1.5–1.5 | 1.5–1.7 | 1.7–1.8 |
| **Spiracle distal edge height** | 4.2–4.3 | 3.9–4.1 | 3.6–4.6 | 3.8–4.9 |
| **Snout-spiracular distance** | 10.0–10.4 | 9.9–10.2 | 9.8–9.8 | 9.4–10.2 |
| **Eye diameter** | 1.8–1.9 | 1.9–1.9 | 1.9–1.9 | 1.8–1.9 |
| **Body width** | 9.3–10.0 | 9.4–9.6 | 8.9–9.3 | 9.4–10.1 |
| **Body width at narial level** | 5.7–6.0 | 5.5–5.7 | 5.6–5.8 | 6.5–6.7 |
| **Body width at eye level** | 7.7–7.8 | 7.6–7.9 | 7.3–7.4 | 7.8–8.3 |
| **Tail muscle width** | 4.5–4.7 | 4.5–4.5 | 4.4–4.4 | 4.6–4.8 |
| **Eye-nostril distance** | 2.4–2.5 | 2.4–2.4 | 2.4–2.5 | 2.5–2.5 |
| **Eye-snout distance** | 4.4–4.8 | 4.4–4.4 | 4.3–4.9 | 4.9–5.1 |
| **Nostril-snout distance** | 2.1–2.4 | 2.0–2.1 | 2.0–2.5 | 2.4–2.5 |
| **Narial diameter** | 0.5–0.6 | 0.5–0.5 | 0.6–0.6 | 0.5–0.6 |
| **Internarial distance** | 2.8–2.9 | 2.9–3.0 | 2.7–2.8 | 3.1–3.1 |
| **Interorbital distance** | 5.2–5.3 | 5.3–5.3 | 5.1–5.4 | 5.4–5.6 |
| **Oral disc width** | 3.4–3.5 | 3.7–3.8 | 3.5–3.9 | 3.9–4.4 |
| **Vent tube length** | 1.7–2.1 | 2.2–2.5 | 1.4–1.9 | 2.1–2.3 |
| **Oral-disc position** | 13.2˚–18.3˚ | 7.6˚–9.4˚ | 14.2˚–21.8 | 21.4˚–24.5˚ |
| **Dorsal-fin insertion angle** | 12.6˚–14.4˚ | 12.7˚–18.5˚ | 9.2˚–22.1˚ | 17.3˚–22.3˚ |

midway between eyes and the tip of snout (NSD/ESD = 0.46–0.51); presence of small fleshy projection on medial margin; lateral margin slightly scalloped (Fig 6D and 6E). Eyes medium-sized (ED/BWE = 0.23–0.26), dorsally located (IOD/BWE = 0.67–0.74), slightly anterolaterally directed. Spiracle sinistral, lateral, visible in dorsal and ventral views (SDH/BH = 0.57–0.62), short (SL/BL = 0.10–0.12), posterodorsally projected; inner wall free from the body and almost the same length as the external wall; opening elliptical, narrower than the anterior portion of the spiracular tube, located between the medial and posterior thirds of the body (SSD/BL = 0.64–0.71; Fig 6B and 6C). Vent tube moderately long (VTL/BL = 0.10–0.17), dextral; ventral wall fused to ventral fin and longer than dorsal wall (Fig 6F). Intestinal tube visible due to the transparency of the abdominal wall, circularly coiled, switchback point somewhat dislo-cated to the left of the abdominal cavity. Tail long (TAL/TL = 0.71–0.72), low (MTH/TAL = 0.15–0.19), slightly higher than body (MTH/BH = 1.02–1.28); tail musculature robust (TMH/BH = 0.64–0.71) reaching tip of pointed tail. Dorsal and ventral fins low (DFH/TAL = 0.08–0.09; VFH/TAL = 0.06–0.07), with slightly convex free margins; dorsal fin emerg-ing on posterior third of the body at a low slope (DFiA = 9˚–28˚); maximum height at poste-rior third of tail; ventral fin origin at level of vent tube. Oral disc (Figs 5C and 6A) ventrally positioned (ODP = 7.6˚–24.5˚), large (ODW/BW = 0.35–0.43, measured with oral disc closed);

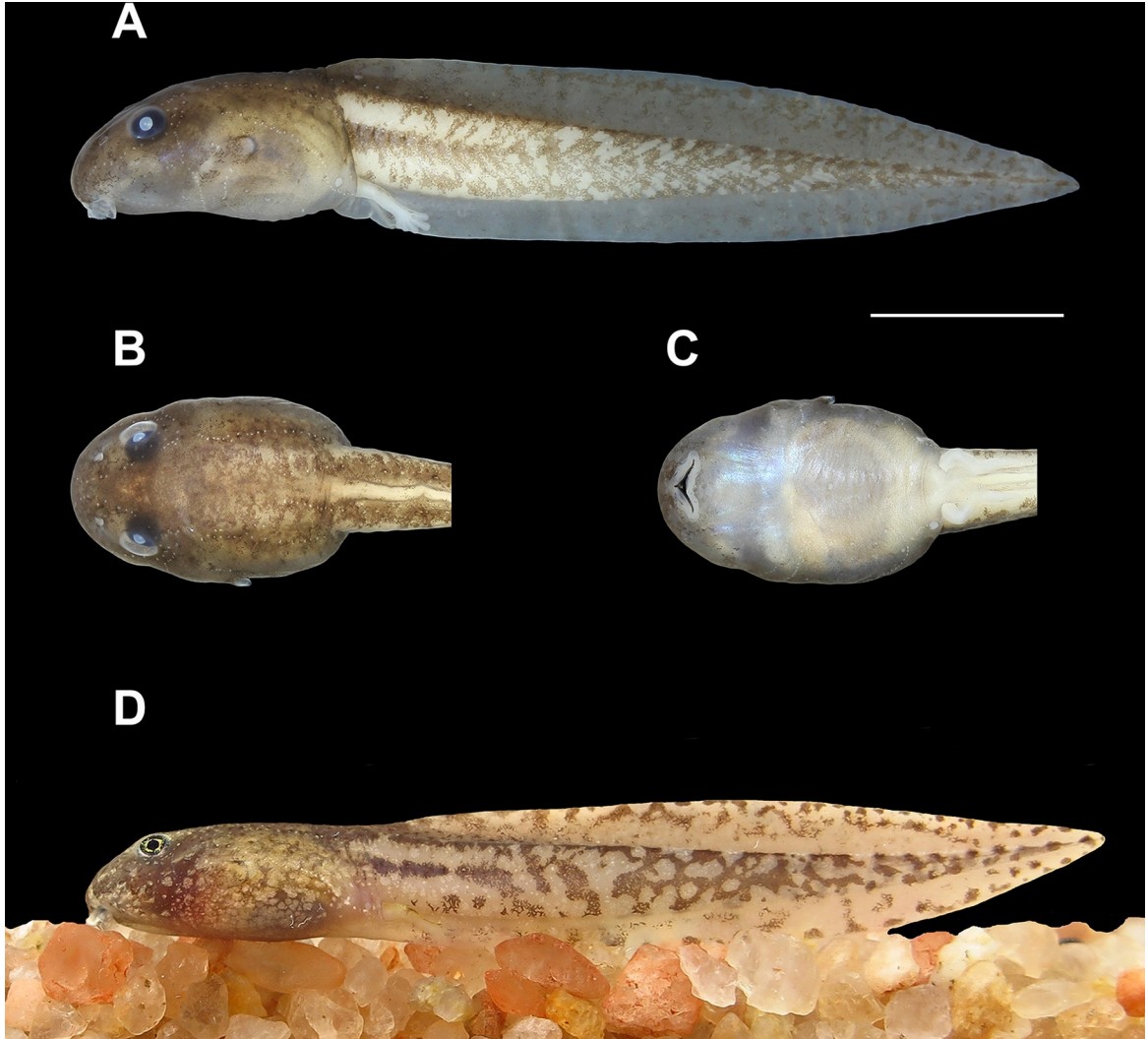

**Fig 5. Tadpole of *Bokermannohyla astartea*.** (A) Lateral, (B) dorsal, and (C) ventral views (lot CFBH 42649) in stage 36, total length 51.5 mm. Scale bar = 10 mm. (D) Tadpole in life (lot CFBH 42649) in stage 35, total length 50.7 mm.

when closed, posterior margin has three emarginations (one medial and two lateral) and the infra-angular region is smoothly directed posteriorly; a single row of conical and rounded marginal papillae (165–180 papillae surrounding the entire disc) interrupted by a moderately wide anterior gap (AGL/ODW = 0.15–0.16; Figs 5C and 6A); papillae aligned in anterior portion of oral disc and with bases offset on lateral and posterior portions. Absence of submarginal papillae and flaps with labial teeth laterally in the oral disc. Labial tooth row formula 2 (2)/4(1); A1 and A2 approximately equal in length; P1 slightly shorter than P2, P2 longer than P3, which is longer than P4; gaps in A2 and P1 about 5 and 1% of the oral disc width, respectively; tooth density on P1 36–45 teeth/mm. Jaw sheaths wide, pigmented, finely serrated on margins (42–50 triangular serrations on upper jaw sheath); upper jaw sheath arc-shaped and lower jaw-sheath V-shaped; lateral processes of anterior jaw sheath medially directed when oral disc is opened. Lateral line system evident in preserved and live specimens (Figs 5A, 5B and 5D). Ten lines can be differentiated, all with rounded stitches. In dorsal view, supraorbital

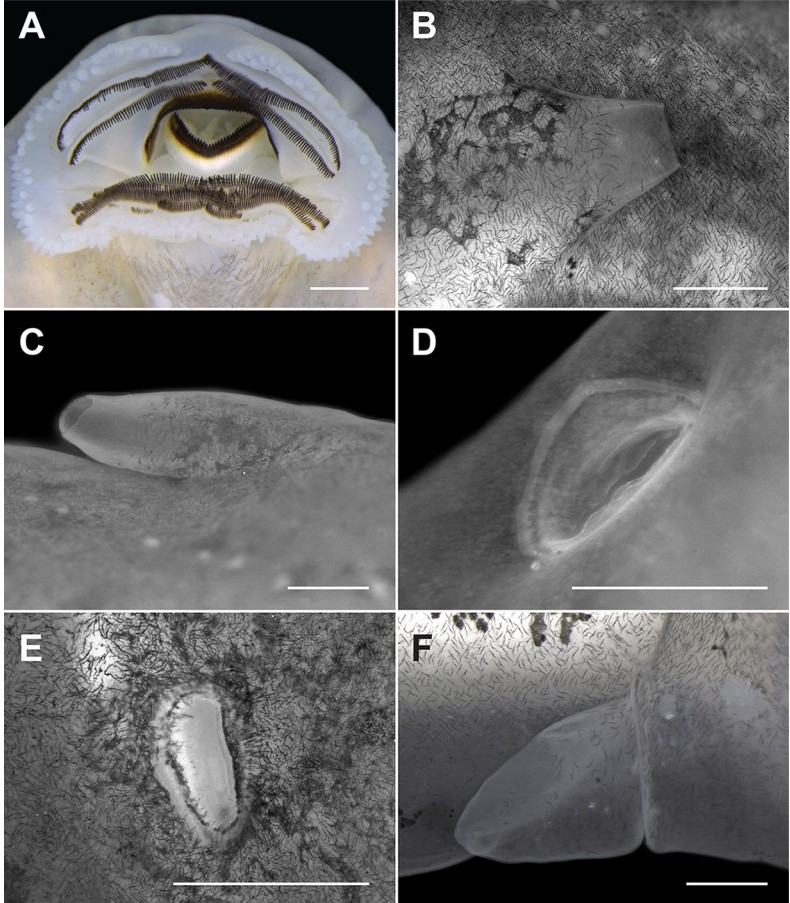

**Fig 6. Details of some external characters of the tadpole of *Bokermannohyla astartea* (lot CFBH 42649) in stage 36.** (A) Typical oral disc, (B) spiracle in lateral and (C) dorsal views, showing that the external wall is almost the same length as the internal wall. (D) Nostril in lateral view showing the fleshy projection on the medial margin, (E) nostril in dorsal view showing the irregular lateral margin. (F) Vent tube. Scale bars = 5 mm. Measurements of external characters are presented in Table 2.

line, with 17–21 irregularly spaced stitches, runs from posterior region of eyes to converge at the level of nares, then diverges in direction towards anterior portion of oral disc. The posterior infraorbital series consists of three or four stitches. Posterior supraorbital line composed of 5–8 stitches located near posterior portion of supraorbital line. Two series of lines extend from mid-body to tail: (1) dorsal line begins with two to four stitches at mid body; after a large gap without neuromasts, 14–20 stitches are distributed from the region of dorsal fin insertion until end of first third of tail; tail stitches are distributed along base of dorsal fin; (2) middle-body line of 19–21 stitches located laterally and joins middle caudal series; this line curves slightly upward along approximate middle third of tail reaching base of dorsal fin; stitches are barely distinct from middle portion of the tail to tail terminus. In lateral view, infraorbital line consists of 19–22 stitches, which are smaller and more concentrated near anterior region of oral disc. Anterior oral line of 20–24 stitches extends from lateral region of oral disc to venter, where it approaches the angular line. Angular line, one of the most noticeable series, has 21–30 stitches and extends obliquely from below eye to venter. Longitudinal oral line of seven or eight stitches extends towards angular line, parallel to infraorbital line. Ventral body line of

31–44 stitches extends from near vent tube to above spiracle; after two gaps without neuromasts at the level of the spiracle, this line continues posteroventrally until mid-abdomin. Well-defined, rounded, and large unpigmented paired spots (*sensu* Quinzio & Goldberg [77]) present above ventral line anterolaterally to base of vent tube (Fig 5A and 5C).

**Coloration.**    In life, general color dark to light brown (Fig 5D). Body marbled by golden spots and black dots, which are scattered mainly over the dorsum; abdominal region whitish; distal margin of spiracle lightly pigmented. Lateral line system highlighted by chromatophores. Iris black and almost completely covered by golden spots, which form a golden rim surrounding the pupil. Tail cream-colored, with musculature and fins mottled by irregular dark brown spots; a short longitudinal dark stripe extends along the medial caudal muscle line, for the first anterior 1/4 of the tail; dorsal margin of caudal musculature with a continuous or interrupted narrow brown line; dorsal fin more densely marbled than ventral fin. Coloration in preservative is similar to that of live individuals but faded and without golden pigmentation of the iris; the tail fins become transparent.

**Variation.**    Individuals from other lots did not exhibit considerable differences in morphology when compared to the analyzed specimens (lots CFBH 42649, 42650, 42651, 42652; Table 2). Four tadpoles in stages 32 (1), 34 (2), and 36 (1) had a longer tail (BL/TL = 0.25–0.27; lot CFBH 38055). Two specimens in stage 35 (lot CFBH 42652), one in stage 36 (lot CFBH 42655), one in stage 25 and one in stage 26 (lot CFBH 42659), had few submarginal papillae (2–4) scattered laterally in the oral disc. The number of stitches was highly variable among specimens of all lots. The ventral body-line was the most variable with 30–55 stitches (lot CFBH 42649). A variable number of well-defined whitish spots (*i.e.*, 1–6), like those commonly found anterolaterally at the base of vent tube, were found scattered on the venter of seven specimens in stages 34 (2), 36 (1), lot CFBH 38055; stage 35 (1), lot CFBH 42649; and stages 36 (2), and 27 (1), lot CFBH 42650. One specimen in stage 35 (lot CFBH 42649; Fig 5D) and two in stage 25 (lot CFBH 42659) had the dorsal fin emerging posteriorly to the body-tail junction at a very low slope (0˚–2˚).

**Comparisons with other species of *Bokermannohyla*.**    As for most species of the *B. circumdata* group, tadpoles of *B. astartea* can be promptly distinguished from species of the *B. pseudopseudis*, and *B. martinsi* groups by the absence of small flaps with labial teeth laterally in the oral disc and the absence of submarginal papillae as the main condition [40, 78]. Besides these character states, tadpoles of *B. astartea* are similar to many species of the *B. circumdata* group in having a general color pattern of brown with dark longitudinal stripes on the caudal musculature [78]. Within the *B. circumdata* group, the tadpoles of *B. astartea* differ from those of *B. diamantina*, *B. gouveai*, and *B. luctuosa* by having four posterior tooth rows (three posterior rows in those three species; [78–80]). This character also distinguishes *B. astartea* from *B. carvalhoi* and *B. nanuzae*, which have five posterior rows as the most common condition [78, 81, 82]. The tadpoles of *B. astartea* differ from other species of the *B. circumdata* group by possessing slightly anterolaterally directed eyes and the external wall of spiracle being almost the same length as the internal wall (eyes dorsolaterally directed and internal wall of spiracle longer than the external wall in the other species; [78–88]).

## Vocal repertoire

The vocal repertoire of *Bokermannohyla astartea* is composed of at least four types of calls, which were categorized based on field observations as: (1) advertisement (n = 11 males); (2) courtship (n = 2 males); (3) amplectant (n = 1 male); and (4) presumed territorial (n = 1 male). Advertisement calls are composed of two types of notes, herein referred to as Notes A and B, respectively. Courtship calls are similar to advertisement calls. Amplectant calls are composed of two short

**Table 3. Acoustic parameters of calls of *Bokermannohyla astartea*, except for courtship call.**

| Call traits | Advertisement call (n = 11 males; 101 calls) | | Amplectant call (n = 1 male; 3 calls; 6 notes) | Presumed territorial call (n = 1 male; 10 calls) |
|---|---|---|---|---|
| | Note A (n = 102 series; 194 notes) | Note B (n = 26 series; 115 notes) | | |
| **Call duration (milliseconds)** | 1888 ± 1607 | | 392 ± 47 (352–444) | 864 ± 423 (233–1529) |
| | (194–17,440) | | | |
| | In series = | In series = | | |
| | 669 ± 160 | 6,206 ± 8,545 | | |
| | (194–1870) | (916–16,646) | | |
| **Number of notes per series/ call** | Single or in series of 2–4 | Single or in series of 2–27 | 2 | Varies according to emission pattern and acoustic component (*e.g.*, note, higher pitched pulse or trill); see text for details. |
| **Note duration (milliseconds)** | 248 ± 44 | 430 ± 70 | 50 ± 17 | |
| | (65–409) | (270–630) | (30–70) | |
| **Number of pulses per note** | 29 ± 7 | 45 ± 11 | 9 ± 1 | |
| | (12–56) | (27–69) | (8–10) | |
| **Pulse rate (pulses/ second)** | 119.1 ± 24.5 | 105.2 ± 19.3 | 202.7 ± 63.4 | |
| | (66.0–246.1) | (66.4–177.2) | (142.8–300.0) | |
| **Dominant frequency (Hz)** | 1876 ± 213 | 1864 ± 229 | 1344 ± 24 | |
| | (1312–2484) | (1265–2484) | (1312–1359) | |
| **Call rate** | 22.6 ± 8.2 (10.5–36.4) (n = 12) | | 44.3 | 21.4 ± 5.5 |
| **(/min)** | | | | (17.5–25.2) (n = 2) |
| **Water/air temperature (˚C)** | 20.2 ± 1.1 (19.0–20.9) / 19.8 ± 2.5 (14.9–22.3) | | 18.7 air | 18 air |

Data are presented as mean ± standard deviation (minimum–maximum). See text for details.

multipulsed notes. A fourth type of call that is presumed to have a territorial function has a distinct structure compared to the other types. Descriptive statistics are given in Table 3 and additional information is provided in S1 Table. Detailed description of each call follows:

**Advertisement call.** Advertisement call duration (considering both Note A and Note B) is 1888 ± 1607 ms (194–17,440, n = 101 calls) and call rate (considering both note types) is 22.6 ± 8.2 calls/min (10.5–36.4; n = 12).

Note A is multipulsed and emitted singly (n = 24 cases of single emission) or in series of 2–4 notes (Table 3; Fig 7A; n = 102 series), each of which has a progressive ascending amplitude with discernible pulses for two-thirds or half of the note, reaching a peak (rise-time of 67 ± 7% [37–88; n = 194 notes]) and decaying towards the end of the note; pulses are juxtaposed in the final portion. Note duration is 248 ± 44 ms (65–409, n = 194 notes), with a dominant frequency of 1876 ± 213 Hz (1312–2484). Notes have 29 ± 7 (12–56) pulses, which are emitted at rates of 119.1 ± 24.5 pulses/second (66.0–246.1). The interval between Notes A is 165 ± 42 ms (100–582; n = 113 intervals).

Note B is facultatively emitted singly (n = 39 cases) or in series (Table 3; Fig 7A; n = 26 series), always shortly after Note A. Note B is longer than Note A and the amplitude peak is usually at the first-third of the note (rise-time of 40 ± 17% [13–95; n = 115 notes]), subtly decaying until the end. Duration of Note B is 430 ± 70 ms (270–630, n = 115 notes), with a dominant frequency of 1864 ± 229 Hz (1265–2484). Notes have 45 ± 11 (27–69) pulses, which are emitted at rates of 105.2 ± 19.3 pulses/seconds (66.4–177.2). Pulses within notes are discernible and always arranged in 4 to 8 pulse groupings. Males could add up to 27 notes in a row. The interval between Notes B is 265 ± 209 ms (106–613; n = 43 intervals). The interval between Note A and Note B is 189 ± 76 ms (103–648; n = 79 intervals).

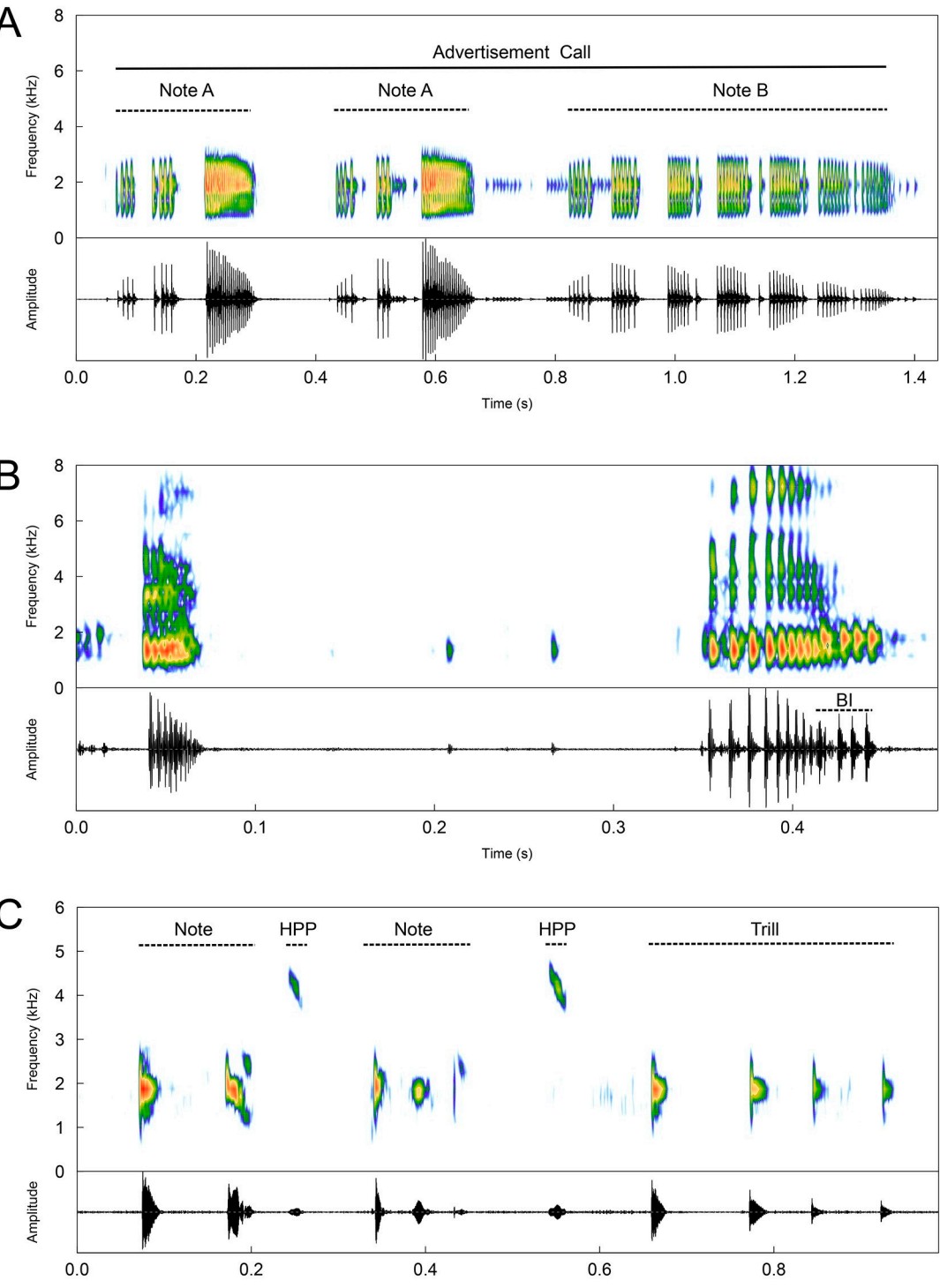

**Fig 7. Vocal repertoire of *Bokermannohyla astartea*, except for courtship call.** Spectrogram (top panels, in color) and respective oscillograms (lower panels, in black) of: (A) an advertisement call depicting the two note types: Note A and Note B (sound file FNJV 45386; voucher specimen CFBH 38044; recorded on 14 October 2014, 1942 h, air temperature of 20.6˚C); (B) an amplectant call with two notes (BI = background uncategorized call from another individual; sound file FNJV 45388; voucher specimen CFBH 38044; recorded on 14 October 2014, 2350 h, air temperature of 18.7˚C); (C) a presumed territorial call depicting typical notes, higher pitched pulses (HPP), and a trill (sound file FNJV 45384; voucher specimen CFBH 42057; recorded on 28 December 2016, 2015 h, air temperature of 22˚C). See text for details.

**Courtship call.** This call was emitted by males when they detected nearby females in bromeliad leaf-tanks and has identical structure to advertisement calls, with emissions of both Notes A and B; however, with lower volume intensity than the advertisement call (Malagoli, L. R., personal observation).

**Amplectant call.** This call (Table 3; Fig 7B; n = 3 calls) was recorded when a male entered into amplexus with a female. Call duration is of 392 ± 47 ms (352–444) with the interval between calls ranging from 967–1269 ms (n = 2 intervals). Each call has two short multipulsed notes, each with a duration of 50 ± 17 ms (30–70; n = 6 notes), 9 ± 1 (8–10) pulses, which are emitted at rates of 202.7 ± 63.4 pulses/second (142.8–300.0) and dominant frequency of 1344 ± 24 Hz (1312–1359). The interval between notes is 291 ± 28 ms (268–323; n = 3 intervals). Call rate is 44.3 calls/min.

**Presumed territorial call.** We observed a resident male (CFBH 42057, SVL 40.18 mm) emitting a conspicuous vocalization in the presence of a satellite male (CFBH 42056, SVL 39.12 mm). The distance between the resident and satellite male was 100 cm. This call (Table 3; Fig 7C; n = 10 calls) had duration of 864 ± 423 ms (233–1529). It shows at least three acoustic units: (1) typical harmonic notes containing two to four pulses, which can have either positive or negative frequency modulations; (2) a higher pitched pulse with downward frequency modulation that is emitted shortly after a typical note; and (3) a trill of three to four pulses with steady frequency throughout its duration. Emission patterns observed were: (1) notes alone; (2) notes + higher pitched pulse; (3) one or two repetitions of the second pattern; (4) two repetitions of (notes + higher pitched pulse) + one or two trills; and (5) note + higher pitched pulse + a single pulse similar to that of notes + higher pitched pulse. All of these patterns are perceived as multiple repetitions of clicks similar to the sound of rainfall drops. Note duration is 135 ± 44 ms (64–267; n = 19 notes) and the dominant frequency is 1931 ± 143 Hz (1641–2297; n = 19 notes). Each pulse within a note has a frequency modulation of -750–234 Hz (n = 53 pulses). Higher pitched pulses have a dominant frequency of 4130 ± 136 Hz (3843–4406; n = 20 pulses) and negative modulation from -328–-47 Hz. Trills have duration of 256 ± 17 ms (234–276; n = 4 trills) and a dominant frequency of 1887 ± 59 Hz (1828–1969). The interval between calls (considering all units) is 2400 ± 2336 ms (552–8413; n = 10 intervals) and call rate is 21.4 ± 5.5 calls/min (17.5–25.2; n = 2).

## Discussion

### Temporal breeding pattern and habitat use

The use of bromeliads as vocalization sites is known for several species of other genera within Cophomantini (see [89–93]), including other species of *Bokermannohyla* [33, 44, 94]. However, the use of bromeliads for breeding (spawning and development of tadpoles) has been recorded only for *Boana pardalis*, although it was interpreted as a case of behavioral plasticity in relation to its most common reproductive mode (*i.e.*, oviposition in natural or constructed basins, reproductive mode 4 [10, 95]). Despite these records, we found that *B. astartea* is dependent on bromeliads for egg deposition and partial development of tadpoles. Deposition of eggs in a microenvironment other than bromeliad leaf-tanks was not observed for *B. astartea*. Therefore, *B. astartea* represents the only known bromeligenous species within the tribe Cophomantini thus far. The reproductive period recorded for *B. astartea* in NC coincides with that observed for populations at Estação Biológica de Boracéia [96]. In fact, both the reproductive period of *B. astartea* (spring and summer) and the area of its occurrence, *i.e.*, Serra do Mar [31], present the highest annual precipitation rate in the Atlantic Forest [97], which may favor the prolonged breeding season of the species and the use of bromeliads to reproduce.

Males and females of *B. astartea* differed in SVL, with females being larger than males. Although this is a relatively common pattern among anurans [98], it has not been observed for many other species of *Bokermannohyla* in different species groups, such as *B. caramaschii* [99], *B. ibitiguara* [37], *B. nanuzae* [39], *B. oxente* [100], and *B. napoli* [101], to cite few. Fecundity in anurans is often positively correlated with female size and number and/or size of eggs [9, 27, 102, 103]. However, the factors responsible for *B. astartea* females being larger than males still need to be evaluated.

## Male site fidelity, satellite behavior, and territoriality

Fidelity to sets of bromeliads was observed for resident males of *Bokermannohyla astartea*. Territoriality in anurans is mainly related to competition for reproductive resources [60, 61]. Therefore, our observations indicate that the maintenance of a set of bromeliads may be advantageous, since the calling site is the same site used for courtship, amplexus, and oviposition. Although we did not observe fights, we think it is likely that they occur since males were observed with scars probably from the prepolical spines of competitor males. For *Bokermannohyla*, male-male combat has only been reported for *B. ibitiguara* [95] and *B. martinsi* [104]. No differences were found between the SVL of resident and satellite males of *B. astartea*, as observed for *B. ibitiguara* [37]. In two cases, a satellite male of *B. astartea* switched to a resident posture and started calling when the resident male was not observed, as also reported for *B. ibitiguara* [37]. However, according to our observations, the satellite behavior of males of *B. astartea* seems to be related to both waiting for calling site vacancy and perhaps opportunistically obtaining females as a sexual parasite of the amplexus of the dominant male (as observed in courtship 2). Similar behaviors have been observed for other hylids, such as *Dendropsophus minutus* and *D. werneri*, indicating that the two reproductive tactics, waiting for calling site vacancy and sexual parasitism, are advantageous for accessing females [55, 105]. The presence of males observing and following amplectant pairs and sitting on newly placed spawn, indicates the possibility of post-mating clutch piracy—*i.e.*, when opportunistic males try to fertilize oocytes released by the amplectant female, which were not fertilized by the resident male (*sensu* Vieites *et al.* [106]) (see Fig 3E and 3F and S2 Movie). Additional studies are required to determine if clutch piracy is employed by *B. astartea*.

## Courtship and oviposition behaviors

Among species of *Bokermannohyla*, courtship behavior has only been reported for *B. alvarengai, B. ibitiguara, B. luctuosa,* and *B. nanuzae*. These four species exhibit complex and elaborate courtship behaviors that include vocalizations, tactile contact, visual signals, as well as males conducting females to the oviposition site (see [36–39]). We did not observe this complexity in the courtship behavior of *B. astartea*, probably because calling and oviposition sites are the same (*i.e.*, leaf-tanks of bromeliads) and thus males are exempted from conducting females to a distant spawning location. On the other hand, females seem to inspect and select leaf-tanks prior to amplexus, possibly evaluating the quality of oviposition sites. Females of *B. ibitiguara* also inspect the oviposition site [37] as do other species of Cophomantini such as some species of *Aplastodiscus* and *Boana* (*e.g.*, [107–109]). Thus, the courtship interruption by females in courtships 1 and 5 may have been because the leaf-tanks evaluated by females did not have adequate conditions for oviposition, such as low water levels or excess of detritus. Rejection of potential oviposition sites after inspection by females has also been reported for *B. ibitiguara* [37] and *Aplastodiscus leucopygius* [107, 108].

We recorded the partitioning of spawns among different leaf-tanks by amplectant females of *B. astartea*. Such partitioning might occur because of the large number of oocytes in the ovaries of females (maximum of 247 oocytes), considering the limited space available in bromeliad

leaf-tanks. Partitioning of spawn may be related to the suitability of oviposition sites (*i.e.*, size of the oviposition site and water availability), as well as the ability to physiologically control oviposition by females and sperm release by males, thereby allowing fertilization in different amplexi [110, 111]. Spawn partitioning has been reported for four species within Hylidae: *Dendropsophus haddadi* [112], a species of the *Scinax perpusillus* group [110], *Pithecopus nordestinus* [111], and *Bokermannohyla luctosa*, although for the last it was performed at the same oviposition site [38]. Thus, *B. astartea* represents the fifth case of spawn partitioning for the family. Considering that this behavior is exhibited by different hylid genera, it is likely to be more common than reported.

## Reproductive mode, spawning, and tadpoles in bromeliads and in streams

The reproductive mode of *Bokermannohyla astartea* is novel for anurans. Instead of having feeding tadpoles that complete their development in bromeliads, which would fit reproductive mode 6 (*sensu* Haddad & Prado [9]), no tadpoles of *B. astartea* beyond stage 26 were found inside bromeliads. Tadpoles beyond this stage were only observed and collected in backwaters of permanent streams or in puddles of temporary streams. Recently, 172 eggs of *B. astartea* were collected in different leaf-tanks of the same bromeliad from Estação Biológica de Boracéia, many of them unfertilized. Among the eggs was a tadpole at stage 25 (CFBH 39948). The bromeliad was an epiphyte located 1.5 m above the backwater of a stream (Baêta, D., personal communication). These data, obtained from a locality about 100-km straight-line-distance from the sites of the present study, reinforce our observations in relation to the use of bromeliads as developmental site to the initial stages of the tadpoles. The oviposition and initial development of tadpoles in bromeliad leaf-tanks exhibited by *B. astartea* suggests similarities with the sites used for spawning and initial tadpole development of other species of the *B. circumdata* group, such as *B. circumdata*, *B. hylax*, *B. luctuosa*, and *B. nanuzae* (see [39, 79, 113]; Malagoli, L.R., personal observation). These species deposit their eggs in small depressions in mud, rock crevices, or under fallen trunks and branches, always near the backwaters of permanent or temporary streams, where the initial development of tadpoles also occurs (see [39, 113]; Haddad, C.F.B., Malagoli, L.R., personal observation). After heavy rains and flooding of these small spaces, tadpoles are transported to the larger water bodies, where they complete development [39, 113]. Likewise, *B. astartea* also oviposit and partitions its spawn in small spaces containing water where tadpoles begin their development (*i.e.*, leaf-tanks of bromeliads), reaching a maximum of 33.6 mm of TL at stage 26 (29.2 ± 2.5 mm, 26.6–33.6 mm, n = 7 tadpoles in stages 25–26, lot CFBH 42659). The tadpoles of *B. astartea* found in streams were at more advanced stages and had greater TL than those found in bromeliads (51.7 ± 0.8 mm, 50.7–52.8 mm, n = 8 tadpoles in stages 35–37; see Table 2), which likely preclude them from completing metamorphosis in bromeliad leaf-tanks. Limited resources in these bromeliad leaf-tanks (*i.e.*, space for development and/or obtaining food; [7]) may favor the exit of tadpoles from leaf-tanks in their earlier developmental stages. Considering the location and position of the bromeliads, always on the margins or above permanent/temporary streams, heavy rains that flood the bromeliads can contribute to the tadpoles presumably falling or jumping into the stream/pond located below or beside bromeliads. Although we did not collect tadpoles of *B. astartea* in the streams beyond March, it is almost certain that they occur beyond this month because we observed some tadpoles in bromeliads in April.

## Tadpole morphology

External larval morphology is currently known for 25 of the 30 species of *Bokermannohyla* [40–42; present study]. The tadpole of *B. astartea* has an overall generalized anatomy

resembling other stream or pond dwelling species of the *B. circumdata* group, including an elongated tail, keratinized mouthparts, and similar LTRF [40–42]. It does not show any reduction of the internal oral features or musculoskeletal specializations (Pezzuti, T.L., personal observation), encountered in highly modified arboreal tadpoles (arboreal tadpoles of Group 1 of Lannoo *et al.* [114]). Thus, the tadpole of *B. astartea* can be classified as belonging to Group III of arboreal tadpoles (*i.e.*, elongate tadpoles with increased denticle rows [114]), with the unique condition of temporarily using bromeliads. Similarly, arboreal tadpoles of Neotropical bufonids of the genus *Melanophryniscus* Gallardo, 1961, did not present marked morphological differences in relation to non-arboreal close relatives that inhabit lentic and lotic environments [20], suggesting few specializations for arboreality when compared to other typical arboreal tadpoles (see [20, 114]). Therefore, factors other than adaptations to specific microhabitats can influence the morphology of tadpoles, such as adaptations to predation/locomotion in different aquatic environments, foraging position in the water column, and phylogenetic relationships [20, 21]. Thus, comprehensive studies are needed to evaluate the importance of tadpole traits both in the reproductive mode of *B. astartea* and within the genus *Bokermannohyla* and the tribe Cophomantini.

## Vocal repertoire

The advertisement call of *Bokermannohyla astartea* was first described by Heyer *et al.* [44], as *Hyla astartea*. Calls in the oscillogram and spectrogram provided therein are represented by three multipulsed notes, corresponding to the Note A series described in the present study. Furthermore, values for call traits including call duration (= note series), number of notes, note rate, number of pulses, and dominant frequency were all within the ranges provided here (Table 3). Heyer *et al.* [44], however, did not have access to Note B or the other call types we describe herein, probably because of the small sample size of calls they had available.

The vocal repertoire of *Bokermannohyla astartea* consists of at least four types of calls, each with very distinct acoustic structures and depending on social context. Advertisement calls of *B. astartea* are composed of two types of notes: Note A, which is emitted in series of one to four similar multipulsed notes; and Note B, also multipulsed, but longer and with a different temporal envelope (*i.e.*, note rise-time; Fig 7A). In some instances, Note B notes can be emitted in extensive series, probably to make competing males more conspicuous to females [7]. This pattern was thus not categorized as a courtship call (as in Zornosa-Torres & Toledo [38]) since they were not emitted during close-range interactions between males and females. Courtship calls emitted in the presence of females by males of *B. astartea* were actually identical to advertisement calls, although with lower intensity, probably following a general trend in anurans in which males drop the intensity of courtship calls to avoid detection by competitors and predators [7]. The extended vocal repertoire of *B. astartea* includes an amplectant call, which is the first reported for the genus. We also report a fourth type of call which we categorized as having a presumably territorial function due to the context in which it was emitted. This call shows an intriguing pattern, with some degree of "acrobatics" by escalating two-fold the pitch of some pulses in a short time span, compared to typical pulses of this call, and by modulating frequencies both downward and upward. This pattern resembles no other pattern observed for species of *Bokermannohyla* and should be addressed in future studies to clearly determine how and why males of *B. astartea* produce such calls.

## Remarks

Just over half a century after the description of *Bokermannohyla astartea* it was possible to obtain detailed information on its natural history, highlighted by unique and unexpected traits for the genus. Such information increases the knowledge on reproduction aspects of species of

*Bokermannohyla*, of which only five out of 30 now have data available ([36–39]; present study). Such data are important not only for a better understanding of how species interact with their environments but also because they provide evolutionary insights on the diversification of reproductive modes in anurans.

Our observations raise the possibility that the reproductive mode of *B. astartea* represents an extreme of the tendency observed for the *B. circumdata* group to spawn outside of main water bodies (*e.g.*, [39, 79, 113]), evolving towards arboreality. In this sense, although *B. astartea* still retains intrinsic characteristics of the group (*e.g.*, large spawn and large detritivorous tadpoles [40, 42, 78]), it possesses a previously unreported mode of tadpole development including a shift in the developmental environment from bromeliads to streams [18]. This shift in tadpole habitat is likely employed to avoid confinement of growing larvae in a small space with limited resources (*i.e.*, oxygen and food) afforded by water accumulated in the bromeliad leaf-tanks [7]. On the other hand, the reasons why adults of *B. astartea* use a microhabitat outside the main water body to reproduce may be related to selective pressures such as aquatic predation of eggs and of very young larvae [15, 16]. Interspecific competition with other Cophomantini, such as the sympatric species *Bokermannohyla circumdata*, *B. hylax*, and *Aplastodiscus* spp., that use microhabitats in margins of streams to breed [44, 107, 108, 113] may also have contributed to the use of bromeliads as a reproductive site by *B. astartea* (see [15, 16, 115]).

Although competition and aquatic predation are hypothesized to be important selective pressures that may lead to terrestriality in anurans (*e.g.*, [3, 4]), some traits of the reproductive biology of *B. astartea* agree with recent complementary hypotheses that are linked to sexual selection [5, 28]. For example, the territoriality of males related to the sets of bromeliads and hidden amplexus and spawning may decrease polyandry [5, 37], which can reduce spawning damage by multiple male harassment [5]. Besides male-male competition, terrestrial reproductive modes have been suggested to influence sexual size dimorphism and fecundity in anurans [28]. Although present, the limited sexual dimorphism in SVL between males and females of *B. astartea* may occur because amplectant females do not carry males to another site for spawning [28]. Still, the space limitation of bromeliad leaf-tanks may favor reduced female size since the pair has to fit in the small space to spawn (*e.g.*, [37, 107]), possibly reducing spawning size in these microhabitats leading to, consequently, less pronounced sexual size dimorphism [5, 28]. These traits are postulated to be strongly related to the evolution and diversification of terrestrial/arboreal reproductive modes in frogs, and reinforces that other selective pressures can act beyond predation and competition [5, 28].

Lastly, the tropical region concentrates more anuran species that have terrestrial reproductive modes than does the temperate region, due to its high precipitation [27, 116]. Considering the present study, 42 reproductive modes are currently recognized for anurans in the world, of which 32 have been described for the Neotropics, 28 of which are represented among species occurring in the Atlantic Forest [9, 11]. Indeed, this morphoclimatic domain shows heterogeneous topographic and climatic characteristics that favor the occurrence of many types of humid microhabitats outside (though linked to) waterbodies. This likely contributed to the diversification of reproductive modes, with many of them showing tendencies towards terrestriality or arboreality [9, 116]. Therefore, despite the current known diversity, our study represents a step towards understanding even more about the basic biology of Neotropical anurans, and reinforces the expectation that novel reproductive modes may be discovered with increased efforts to observe and describe the astonishing aspects of anuran natural history [11, 28].

## Supporting information

**S1 Appendix. Specimens of tadpoles, spawn and adults examined.**
(DOCX)

**S1 Table. Metadata of sound files and types of calls analyzed for *Bokermannohyla astartea*.** All recordings were obtained at Núcleo Curucutu, Parque Estadual da Serra do Mar, state of São Paulo, southeastern Brazil.
(DOCX)

**S2 Table. Tadpole identification with metadata information.** Tissue sample ID, GenBank accession numbers (sequences generated in this study), molecular identification, identity, most similar sequence and respective voucher, lot number of voucher specimen, and collection breeding site and respective environment.
(DOCX)

**S3 Table. Sequential behavioral features of the five observed courtships of *Bokermannohyla astartea*.**
(DOCX)

**S1 Fig. Males of *Bokermannohyla astartea* individualized by their natural dorsal markings.** (A) Male with white blotches on the posterior portion of the dorsum and (B) male with black spots scattered on the dorsum.
(TIF)

**S2 Fig. Male of *Bokermannohyla astartea* (individualized male number 1) showing scratch scars on the dorsum and head.** White arrows indicate some of the scars.
(TIF)

**S1 Movie. Pair of *Bokermannohyla astartea* in amplexus in a bromeliad leaf-tank at breeding site 1.** The male squeezed the female with his arms during axillary amplexus and deposition of eggs. Eighty-two eggs were laid during this amplexus (see text for details).
(MOV)

**S2 Movie. Sequence of reproductive behavior of *Bokermannohyla astartea* at breeding site 1.** The satellite male sits on spawn newly deposited by a pair that at the same moment resume amplexus in the leaf-tank above. The satellite male left the leaf-tank of the spawn and approached the leaf-tank where the pair was in amplexus, watching the pair and passing under the leaf-tank several times, while the resident male emitted an amplexus call. Twenty-six eggs were laid in the last amplexus (see text for details). This is a sequence of behaviors presented in S1 Movie.
(MOV)

**S3 Movie. Final sequence of reproductive behavior of *Bokermannohyla astartea* at breeding site 1.** The female left the resident male, who still tried unsuccessfully to hold her. The resident male returned to the leaf-tank and continued to emit courtship calls. At same time, the satellite male passed occasionally under the leaf-tank where the pair was and takes a crouching posture. When the female deflects the resident male, the satellite male looks back at the female (see text for details). This is a sequence of behaviors presented in S2 Movie.
(MOV)

## Acknowledgments

We thank two anonymous reviewers for their comments and suggestions that improved the manuscript. We thank T. Grant, H. Zaher (MZUSP), J. P. Pombal Jr., M. W. Cardoso, P. Pina (MNRJ), L. F. Toledo, K. Rebelo (ZUEC-AMP) for allowing access to specimens under their care. We thank G. Morales, P. P. G. Taucce, P. D. P. Pinheiro, J. V. A. Lacerda, T. A. L. Oliveira, E. G. Oliveira, F. E. Barbo, R. Guadeluppe, J. M. Onça, P. Z. Soares, and F. Schunck for

helping in the field work. We thank to J. M. Onça, N. C. Puppin, J. P. C. Monteiro and D. P. Baêta for helping in laboratory protocol, with the last also contributing data from Estação Biológica de Boracéia. We thank to V. S. Nascimento, M. Alonso, T. Schmidt, and M. J. Gonçalves for all logistical support at the Núcleo Curucutu from Parque Estadual da Serra do Mar. We thank to B. V. M. Berneck for thoughtful insights in early versions of the manuscript. We thank to C. P. A. Prado for revising an early version of this manuscript. We also thank D. F. dos Santos for obtaining images and measurements of tadpoles. We thank Centro de Estudos de Insetos Sociais (CEIS), UNESP Rio Claro, for use of the molecular laboratory facilities. We thank R. J. F. Garcia from Herbário Municipal de São Paulo (Divisão de Produção e Herbário Municipal/SVMA/PMSP) for identifying bromeliads genera and species. We thank E. Wild and J. M. Onça for improving our use of written English.

## Author Contributions

**Conceptualization:** Leo Ramos Malagoli, Tiago Leite Pezzuti, Célio Fernando Baptista Haddad.

**Data curation:** Leo Ramos Malagoli, Tiago Leite Pezzuti, Davi Lee Bang, Mariana Lúcio Lyra, João Gabriel Ribeiro Giovanelli.

**Formal analysis:** Leo Ramos Malagoli, Tiago Leite Pezzuti, Davi Lee Bang, Mariana Lúcio Lyra, João Gabriel Ribeiro Giovanelli.

**Funding acquisition:** Leo Ramos Malagoli, Ricardo Jannini Sawaya, Célio Fernando Baptista Haddad.

**Investigation:** Leo Ramos Malagoli, Tiago Leite Pezzuti, Davi Lee Bang, Mariana Lúcio Lyra.

**Methodology:** Leo Ramos Malagoli, Tiago Leite Pezzuti, Davi Lee Bang, Mariana Lúcio Lyra, João Gabriel Ribeiro Giovanelli.

**Project administration:** Leo Ramos Malagoli.

**Resources:** Leo Ramos Malagoli, Tiago Leite Pezzuti, Paulo Christiano de Anchietta Garcia, Célio Fernando Baptista Haddad.

**Software:** Leo Ramos Malagoli, Tiago Leite Pezzuti, Davi Lee Bang, Mariana Lúcio Lyra, João Gabriel Ribeiro Giovanelli.

**Supervision:** Julián Faivovich, Paulo Christiano de Anchietta Garcia, Ricardo Jannini Sawaya, Célio Fernando Baptista Haddad.

**Validation:** Leo Ramos Malagoli, Tiago Leite Pezzuti, Davi Lee Bang, Mariana Lúcio Lyra, João Gabriel Ribeiro Giovanelli.

**Visualization:** Leo Ramos Malagoli, Tiago Leite Pezzuti, Davi Lee Bang, Mariana Lúcio Lyra, João Gabriel Ribeiro Giovanelli.

**Writing – original draft:** Leo Ramos Malagoli, Tiago Leite Pezzuti.

**Writing – review & editing:** Leo Ramos Malagoli, Tiago Leite Pezzuti, Davi Lee Bang, Julián Faivovich, Mariana Lúcio Lyra, João Gabriel Ribeiro Giovanelli, Paulo Christiano de Anchietta Garcia, Ricardo Jannini Sawaya, Célio Fernando Baptista Haddad.

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
