## [Decision Letter · Decision Letter 0]

11 Sep 2020

PONE-D-20-19802

A new reproductive mode in anurans: natural history of Bokermannohyla astartea (Anura: Hylidae) with the description of its tadpole and vocal repertoire

PLOS ONE

Dear Dr. Malagoli,

Thank you for submitting your manuscript to PLOS ONE. After careful consideration, we feel that it has merit but does not fully meet PLOS ONE’s publication criteria as it currently stands. Therefore, we invite you to submit a revised version of the manuscript that addresses the points raised during the review process.

There were two referees who both find the results interesting and suggest that they should be published. Also, on the one hand, there are mainly minor comments regarding the manuscript's structure, language etc.

However, on the other hand, one referee made an important point (and therefore even ranked the paper 'reject'), and I share this point. There is a lot of natural history information on a particular species and the focus of the paper is not really strong. That is, the paper as it is would perhaps work well in a herp journal but not so well in a more general science journal. The one major point of your findings, the new reproductive mode, is certainly something special and something beyond a natural history study. But this is not well emphasized given the amount of other information. The referee suggests to split the information into various papers, which I think is a good idea.

Along with this, the introdcution and the discussion remain a bit too general in the paper as it is.

These critics are NOT a reason to reject your paper according to the publication criteria of PLoS ONE. As a result I ranked your paper 'minor revision' and leave the decision with you to perhaps shorten the manuscript for PLoS ONE with a stronger focus on the reproductive mode novelty. 

We look forward to receiving your revised manuscript.

Kind regards,

Stefan Lötters

Academic Editor

PLOS ONE

Journal Requirements:

3. We note that Figure 11 in your submission contain map images which may be copyrighted. All PLOS content is published under the Creative Commons Attribution License (CC BY 4.0), which means that the manuscript, images, and Supporting Information files will be freely available online, and any third party is permitted to access, download, copy, distribute, and use these materials in any way, even commercially, with proper attribution. For these reasons, we cannot publish previously copyrighted maps or satellite images created using proprietary data, such as Google software (Google Maps, Street View, and Earth). For more information, see our copyright guidelines: http://journals.plos.org/plosone/s/licenses-and-copyright.

3.1.    You may seek permission from the original copyright holder of Figure 11 to publish the content specifically under the CC BY 4.0 license. 

3.2.    If you are unable to obtain permission from the original copyright holder to publish these figures under the CC BY 4.0 license or if the copyright holder’s requirements are incompatible with the CC BY 4.0 license, please either i) remove the figure or ii) supply a replacement figure that complies with the CC BY 4.0 license. Please check copyright information on all replacement figures and update the figure caption with source information. If applicable, please specify in the figure caption text when a figure is similar but not identical to the original image and is therefore for illustrative purposes only.

Reviewers' comments:

Reviewer's Responses to Questions

**Comments to the Author**

1. Is the manuscript technically sound, and do the data support the conclusions?

Reviewer #1: Yes

Reviewer #2: Yes

2. Has the statistical analysis been performed appropriately and rigorously? 

Reviewer #1: Yes

Reviewer #2: Yes

3. Have the authors made all data underlying the findings in their manuscript fully available?

Reviewer #1: Yes

Reviewer #2: Yes

4. Is the manuscript presented in an intelligible fashion and written in standard English?

Reviewer #1: Yes

Reviewer #2: Yes

5. Review Comments to the Author

Reviewer #1: This study brings novel and important information on the biology and distribution of a frog species, including the description of a new reproductive mode for anurans. The manuscript is a beautiful natural history study and is well written. Although I recognize the importance of natural history studies to enhance our knowledge on the ecology and evolution of organisms, I have to say that the manuscript suffers from the lack of focus, which makes it extremely long. Additionally, the work is centered in the species biology, which makes it less attractive to a broader audience.

As the study presents diverse aspects of the species biology, I think the authors should consider separating the content of the manuscript in different more focused manuscripts (e.g. reproductive mode and breeding biology; description of tadpole and call types; geographic distribution and conservation). I would like to emphasize that my suggestion is not based solely on the length of the work, but because shorter and more focused studies are more likely to be read and to reach greater impact in the scientific community. Below, I provide some suggestions that might be useful to broaden the focus of the manuscript.

The introduction is very restricted to the taxon studied. It is possible to modify the framework to make the study more interesting to a broader readership. For instance, the great diversity of reproductive modes in frogs could be introduced in the light of the evolution of terrestrial reproduction in vertebrates and the factors leading to terrestriality. Furthermore, other aspects of the study are not presented in the introduction. The importance of acoustic communication and diversity of acoustic repertoires in many animal groups, including anurans, could be mentioned in the introduction. Similarly, as the manuscript uses predictive models to infer geographic distribution and discuss the conservation status of the species, these subjects should be presented in the introduction (e.g. the situation of amphibian conservation in the world and in Brazil, main threats to the group, the current conservation status of the species and the importance of predicting geographic distribution to support conservation policies).

Similarly to the introduction, the discussion section presents a superficial discussion because of the great variety of subjects in the manuscript. For instance, in the first paragraph of the discussion, if water bodies are available, I was wondering about the factors that could have led to the reproduction in bromeliads in this species (e.g. to avoid competition with other species in aquatic sites, avoid predation, avoid competitor males or sneakers?). There are recent papers discussing the evolution of terrestrial reproduction in frogs that could be mentioned. Furthermore, in the last years, new reproductive modes have been described for anurans and currently we have more than 40 reproductive modes (mentioned in the introduction), many of them described for species from tropical forests outside the Neotropics. Factors related to the evolution of this striking diversity of reproductive modes among tropical frogs could also be discussed.

Reviewer #2: The study by Malagoli et al. is a very nice piece of amphibian natural history, well written and richly and beautifully illustrated. It describes a new reproductive mode in anurans, courtship, defensive behaviors, spawning, and tadpoles of Bokermannohyla astartea. They also update the description of its vocal repertoire and provide updated information on its geographic distribution and conservation status. There is some criticism surrounding the description of new reproductive modes in amphibians because very subtle differences in reproductive biology are sometimes used to propose a ´new reproductive mode´. The reproductive mode of B. astartea seems, however, indeed distinctive. Of course, there is an important element missing… that is, how do tadpoles end up in surface waterbodies. Maybe the authors could be more explicit about whether they have invested the time and effort needed to find tadpoles above stage 26 in bromeliads, or tadpoles below stage 26 in streams. Descriptions of behavior, tadpoles and calls (based on my limited experience with bioacoustics) are all very thorough. Again, the pictures and, to be honest, all graphical elements, are excellent. I only have minor comments that could be addressed to improve the paper.

Line 118 - Seventy-one-days exclusively or predominantly devoted to observing B astartea? Or 71 days in which several tasks were conducted in the field including observing B astarea?

Line 119 – Observations were done in only two breeding sites? Is this species of difficult access in other localities within the range of its distribution?

Lines 133 – 145 – Figure titles for figures 1 and 2 are the same. I believe you could change them to better indicate what each figure shows. Alternatively, you could rearrange those two panels into a figure that shows the breeding sites and its differences (temporary vs permanent streams? BS1 vs BS2?) and another showing the position of calling activity on the BSs.

Line 147 - What is a ´all-occurrence sampling method´?

Lines 164-165 - Liver and muscle samples were preserved… What for?

Lines 167 – 170 – How many specimens in each category (resident males, satellite males, females)?

Lines 191, 626, 687 - Presumed? Presumably?

Lines 228 a 244 –It is not absolutely necessary, but It would be helpful for readers to have the measurements of the external morphology indicated in a figure like Figure 6.

Lines 309 – 311 – Did they use 75% of the data to train the algorithms and 25% to validate it? Please, explain in a bit more detail.

Lines 339 – 341 – When you show the number of individuals (n), also provide percentages so we can immediately know how rare or common the behavior was in your sample size. Like you did in “Males vocalizing outside bromeliads (6.7%, n = 6)”.

Line 345 –“ 39.7 ± 1.3 mm (36.8–42.7 mm, n = 40)”. What are these intervals inside the parenthesis? Minimum and maximum values? Confidence intervals?

Line 363 – “(n = 14, % of total males observed?)”

Line 366-369. I don’t think you have to indicate the full study locality in the figure title. Just “Males of Bokermannohyla astartea recaptured in the same sets of 367 bromeliads in the two breeding sites sampled (BS1 and BS2)” would be enough.

Line 399. The male does not spawn. I assume you mean ´after egg laying was completed´ or something like that.

Lines 427-428. Is the satellite male observing or being observed? I believe you meant “no satellite males observing” here.

Line 443. ´ Spawns were always found in bromeliad leaf-tanks´ - as opposed to? All your sampling effort was invested into bromeliads. Would you have found spawns in a surface waterbody?

Line 449 – “(139–194)” minimum and maximum values?

Line 535 – 538. I cannot fully see four posterior tooth rows or the gap in the first tooth row in Figure 7A. This is not necessarily a problem and I know those pictures are hard to obtain. Still, do you have another picture that clearly shows them?

Lines 647 – Again, you do not need to repeat the locality in the title of the table.

Line 752 – Remove comma in “Even, the predicted occurrence”

6. PLOS authors have the option to publish the peer review history of their article (what does this mean?). If published, this will include your full peer review and any attached files.

Reviewer #1: No

Reviewer #2: No

---

## [Author Response · Author response to Decision Letter 0]

8 Nov 2020

Editor:

1.Thank you for submitting your manuscript to PLOS ONE. After careful consideration, we feel that it has merit but does not fully meet PLOS ONE’s publication criteria as it currently stands. Therefore, we invite you to submit a revised version of the manuscript that addresses the points raised during the review process.

There were two referees who both find the results interesting and suggest that they should be published. Also, on the one hand, there are mainly minor comments regarding the manuscript's structure, language etc.

However, on the other hand, one referee made an important point (and therefore even ranked the paper 'reject'), and I share this point. There is a lot of natural history information on a particular species and the focus of the paper is not really strong. That is, the paper as it is would perhaps work well in a herp journal but not so well in a more general science journal. The one major point of your findings, the new reproductive mode, is certainly something special and something beyond a natural history study. But this is not well emphasized given the amount of other information. The referee suggests to split the information into various papers, which I think is a good idea.

Along with this, the introdcution and the discussion remain a bit too general in the paper as it is. These critics are NOT a reason to reject your paper according to the publication criteria of PLoS ONE. As a result I ranked your paper 'minor revision' and leave the decision with you to perhaps shorten the manuscript for PLoS ONE with a stronger focus on the reproductive mode novelty.

Answer: We carefully reviewed each question and suggestions, trying to include them in the revised manuscript. We have reduced the size of the manuscript removing the sections of ecological modeling, distribution and conservation, and also the section about defensive behavior, reducing as well the number of figures from 11 to 7. We have also restructured the Introduction and the Discussion making it more focused on the new reproductive mode, to make it more attractive to broader viewers.

Reviewer #1

1. This study brings novel and important information on the biology and distribution of a frog species, including the description of a new reproductive mode for anurans. The manuscript is a beautiful natural history study and is well written. Although I recognize the importance of natural history studies to enhance our knowledge on the ecology and evolution of organisms, I have to say that the manuscript suffers from the lack of focus, which makes it extremely long. Additionally, the work is centered in the species biology, which makes it less attractive to a broader audience.

As the study presents diverse aspects of the species biology, I think the authors should consider separating the content of the manuscript in different more focused manuscripts (e.g. reproductive mode and breeding biology; description of tadpole and call types; geographic distribution and conservation). I would like to emphasize that my suggestion is not based solely on the length of the work, but because shorter and more focused studies are more likely to be read and to reach greater impact in the scientific community. Below, I provide some suggestions that might be useful to broaden the focus of the manuscript.

Answer: Thank you for the all the valuable suggestions. We have suppressed the sections concerning to defensive behaviors, geographic distribution, and conservation of the species, and this has reduced the text size and consequently the number of figures, from 11 to 7. Therefore, we tried to put focus in the new reproductive mode, which inherently englobes the species reproductive biology. Although the suggestion to separate the content of our work is appreciated, we deemed that data on tadpoles and calls of the species to be of crucial importance to supplement our description of the new reproductive mode. Moreover, we believe that the description of the only bromeligenous tadpole of the Cophomantini tribe, can add more value to the manuscript, considering that the description of the new reproductive mode involves the change in the tadpole's development habitat, implying differences in the stage of development and size of the tadpoles in each environment (i.e., bromeliads and in streams/ponds). In this sense, despite the manuscript is now more focused in the species reproductive biology, we restructured Introduction and Discussion sections in an attempt to reach more broaden audience.

2. The introduction is very restricted to the taxon studied. It is possible to modify the framework to make the study more interesting to a broader readership. For instance, the great diversity of reproductive modes in frogs could be introduced in the light of the evolution of terrestrial reproduction in vertebrates and the factors leading to terrestriality. Furthermore, other aspects of the study are not presented in the introduction. The importance of acoustic communication and diversity of acoustic repertoires in many animal groups, including anurans, could be mentioned in the introduction. Similarly, as the manuscript uses predictive models to infer geographic distribution and discuss the conservation status of the species, these subjects should be presented in the introduction (e.g. the situation of amphibian conservation in the world and in Brazil, main threats to the group, the current conservation status of the species and the importance of predicting geographic distribution to support conservation policies).

Answer: We agree with the suggestions made. We have reformulated the introduction to make it more comprehensive and attractive to broader readership. We present some points related to the evolution of terrestrial reproduction in vertebrates and the factors leading to terrestriality, including anurans. Additionally, we have included information about the importance of describing tadpoles and knowledge about the natural history of larvae in the context of reproductive modes. Likewise, as suggested, we pointed out on the importance of communication and acoustic repertoire for different animal groups, including anurans. With the intention of making our work more focused in the new reproductive mode, we removed the Geographical distribution and conservation section, and respective results and discussion of the manuscript.

3. Similarly to the introduction, the discussion section presents a superficial discussion because of the great variety of subjects in the manuscript. For instance, in the first paragraph of the discussion, if water bodies are available, I was wondering about the factors that could have led to the reproduction in bromeliads in this species (e.g. to avoid competition with other species in aquatic sites, avoid predation, avoid competitor males or sneakers?). There are recent papers discussing the evolution of terrestrial reproduction in frogs that could be mentioned. Furthermore, in the last years, new reproductive modes have been described for anurans and currently we have more than 40 reproductive modes (mentioned in the introduction), many of them described for species from tropical forests outside the Neotropics. Factors related to the evolution of this striking diversity of reproductive modes among tropical frogs could also be discussed.

Answer: We agree with the suggestions made. At the end of the discussion section, we included a “Remarks” section, formerly named “Concluding remarks”. In this section we include all the suggestions made by the Reviewer, addressing the issues in a more broadly sense, trying to discuss the traits of the species' newly reproductive biology in relation with the factors that could have driven the evolution and diversification of terrestrial/arboreal reproductive modes in anurans.

Reviewer #2

1. The study by Malagoli et al. is a very nice piece of amphibian natural history, well written and richly and beautifully illustrated. It describes a new reproductive mode in anurans, courtship, defensive behaviors, spawning, and tadpoles of Bokermannohyla astartea. They also update the description of its vocal repertoire and provide updated information on its geographic distribution and conservation status. There is some criticism surrounding the description of new reproductive modes in amphibians because very subtle differences in reproductive biology are sometimes used to propose a ´new reproductive mode´. The reproductive mode of B. astartea seems, however, indeed distinctive. Of course, there is an important element missing… that is, how do tadpoles end up in surface waterbodies. Maybe the authors could be more explicit about whether they have invested the time and effort needed to find tadpoles above stage 26 in bromeliads, or tadpoles below stage 26 in streams. Descriptions of behavior, tadpoles and calls (based on my limited experience with bioacoustics) are all very thorough. Again, the pictures and, to be honest, all graphical elements, are excellent. I only have minor comments that could be addressed to improve the paper.

Answer: We really appreciate all the suggestions and corrections. We agree with the observation that the way the tadpoles end up in the waterbodies from the bromeliads is still unknown. However, through the information we present about the characteristics of its breeding environment, its reproductive biology, and the description of its tadpole, we highlight that the most likely scenario is that the tadpole jump or is transported after the overflow of leaf-tanks of bromeliads to the main waterbody, considering that the streams and rivulets are always located very close to the sets of bromeliads. In addition, we also better emphasize differences in the developmental stages of tadpoles when they are in bromeliads and when they are in streams in the discussion section (in subsection Reproductive mode, spawning, and tadpoles in bromeliads and in streams), also pointing that tadpoles beyond stage 26 were never found within bromeliads. We emphasized in the methods, in the “spawn, egg, and tadpole” section, that the investment in the search for tadpoles occurred in both bromeliads and streams.

2. Line 118 - Seventy-one-days exclusively or predominantly devoted to observing B. astartea? Or 71 days in which several tasks were conducted in the field including observing B. astarea?

Answer: It was 71 days exclusively to observe B. astartea. We better detailed this information in the text.

3. Line 119 – Observations were done in only two breeding sites? Is this species of difficult access in other localities within the range of its distribution?

Answer: Yes, the observations were done focally in two breeding sites in our work. This species is not easily found within the range of its distribution, occurring in specific environments that are not always accessible and not easily to find in the field. 

4. Lines 133 – 145 – Figure titles for figures 1 and 2 are the same. I believe you could change them to better indicate what each figure shows. Alternatively, you could rearrange those two panels into a figure that shows the breeding sites and its differences (temporary vs permanent streams? BS1 vs BS2?) and another showing the position of calling activity on the BSs.

Answer: We agree with the change of text in the titles of figures 1 and 2. We changed the titles to better indicate what each figure shows as suggested, but we chose to keep the figures in the same arrangement because we consider that they are relevant to illustrate both the types of habitats used by the species as well as the relationship with the new reproductive mode described.

5. Line 147 - What is a ´all-occurrence sampling method´?

Answer: This method records all types of behaviors performed by a species during a given time period of observation. We included this information in the text to make it clearer.

6. Lines 164-165 - Liver and muscle samples were preserved… What for?

Answer: We added the sentence: “for DNA extraction protocols”.

7. Lines 167 – 170 – How many specimens in each category (resident males, satellite males, females)?

Answer: We inform how many specimens were measured in each category in the respective sections of results, both in the “Temporal breeding pattern and habitat use” section and in the “Male site fidelity, satellite behavior, and territoriality” section.

8. Lines 191, 626, 687 - Presumed? Presumably?

Answer: We agree that the correct word is presumably and we corrected it throughout the text.

9. Lines 228 a 244 – It is not absolutely necessary, but It would be helpful for readers to have the measurements of the external morphology indicated in a figure like Figure 6.

Answer: We included the measurements of the total length of the tadpoles in the captions of Figure 5 (previously referred to as Figure 6). For the figure 6 (previously referred to as Figure 7), which presented details of some external characters of the tadpole, we do not include the measurements of each external character in the figure legend, because it is uncommon to show these measurements in the captions of this type of figure. To make the reader go (if desired) to the table where all the measurements are present, we include the following sentence in the caption: “The measurements of external characters are presented in Table 2”. 

10. Lines 309 – 311 – Did they use 75% of the data to train the algorithms and 25% to validate it? Please, explain in a bit more detail.

Answer: We removed the Geographical distribution and conservation section and respective results and discussion of the manuscript.

11. Lines 339 – 341 – When you show the number of individuals (n), also provide percentages so we can immediately know how rare or common the behavior was in your sample size. Like you did in “Males vocalizing outside bromeliads (6.7%, n = 6)”.

Answer: We agree. We included the percentages that were missing in the respective observations.

12. Line 345 –“39.7 ± 1.3 mm (36.8–42.7 mm, n = 40)”. What are these intervals inside the parenthesis? Minimum and maximum values? Confidence intervals?

Answer: In this case the interval within the parenthesis are the minimum and maximum values of SVL. At the end of the methods section, we included an explanatory sentence in relation to the values presented throughout the text: “Descriptive statistics are henceforth shown as mean ± standard deviation, (minimum–maximum (when applicable), and number of individuals or specific records).”.

13. Line 363 – “(n = 14, % of total males observed?)”

Answer: We agree. We added the sentence: “15,7% of total males observed in study”. 

14. Line 366-369. I don’t think you have to indicate the full study locality in the figure title. Just “Males of Bokermannohyla astartea recaptured in the same sets of bromeliads in the two breeding sites sampled (BS1 and BS2)” would be enough.

Answer: We agree. We removed the sentence: “from Núcleo Curucutu, Parque Estadual da Serra do Mar, in the municipalities of São Paulo and Itanhaém, state of São Paulo, Brazil, respectively”.

15. Line 399. The male does not spawn. I assume you mean ´after egg laying was completed´ or something like that.

Answer: We agree. We replaced the term spawning by “egg laying was completed”.

16. Lines 427-428. Is the satellite male observing or being observed? I believe you meant “no satellite males observing” here.

Answer: We apologize. What we meant in this sentence is that in this courtship observation, we did not observe any satellite male. We excluded the word “being” from the sentence. 

17. Line 443. ´ Spawns were always found in bromeliad leaf-tanks´ - as opposed to? All your sampling effort was invested into bromeliads. Would you have found spawns in a surface waterbody?

Answer: Opposed to finding spawns in streams, which were also part of the sampling effort. We complemented the sentence including that the spawns of B. astartea were not found in the streams. We also included and detailed in the methods section the information about the sampling efforts that we invested in both bromeliads leaf-tanks and streams.

18. Line 449 – “(139–194)” minimum and maximum values?

Answer: Yes, minimum and maximum values of mature oocytes. We changed the placement of this range of values to directly after the values of mean and standard deviation, which we think now will be clearer that it refers to minimum and maximum values of mature oocytes.

19. Line 535 – 538. I cannot fully see four posterior tooth rows or the gap in the first tooth row in Figure 7A. This is not necessarily a problem and I know those pictures are hard to obtain. Still, do you have another picture that clearly shows them?

Answer: We appreciate the concern with the improvement of the figure. Unfortunately, this was the only individual photographed, and we could not open the oral disc a bit more without damaging it. However, we think that although it is not completely open, it is possible to count the posterior rows (P4, P3, P2, from the lateral regions, P1 in the medial region) and the gap in P1 can be visualized discretely in the medial region enlarging the figure. We could graphically indicate these structures, but the design was not good, polluting the figure a lot. Thus, we believe that the figure is sufficient to show that there are 4 rows and a gap at P1.

20. Lines 647 – Again, you do not need to repeat the locality in the title of the table.

Answer: Done, we removed the sentence: “from Núcleo Curucutu, Parque Estadual da Serra do Mar, in the municipalities of São Paulo and Itanhaém, southeastern Brazil” from the table caption and all other figures or tables captions that we had included this sentence in the first version.

21. Line 752 – Remove comma in “Even, the predicted occurrence”

Answer: We removed the Geographical distribution and conservation section and respective results and discussion of the manuscript.

---

## [Decision Letter · Decision Letter 1]

22 Dec 2020

PONE-D-20-19802R1

A new reproductive mode in anurans: natural history of Bokermannohyla astartea (Anura: Hylidae) with the description of its tadpole and vocal repertoire

PLOS ONE

Dear Dr. Malagoli,

Thank you for submitting your manuscript to PLOS ONE. After careful consideration, we feel that it has merit but does not fully meet PLOS ONE’s publication criteria as it currently stands. Therefore, we invite you to submit a revised version of the manuscript that addresses the points raised during the review process.

There are some few issues left that need attention. The one referee and myself were very satisifed with your review.

We look forward to receiving your revised manuscript.

Kind regards,

Stefan Lötters

Academic Editor

PLOS ONE

Reviewers' comments:

Reviewer's Responses to Questions

**Comments to the Author**

1. If the authors have adequately addressed your comments raised in a previous round of review and you feel that this manuscript is now acceptable for publication, you may indicate that here to bypass the “Comments to the Author” section, enter your conflict of interest statement in the “Confidential to Editor” section, and submit your "Accept" recommendation.

Reviewer #1: (No Response)

2. Is the manuscript technically sound, and do the data support the conclusions?

Reviewer #1: Yes

3. Has the statistical analysis been performed appropriately and rigorously? 

Reviewer #1: Yes

4. Have the authors made all data underlying the findings in their manuscript fully available?

Reviewer #1: Yes

5. Is the manuscript presented in an intelligible fashion and written in standard English?

Reviewer #1: No

6. Review Comments to the Author

Reviewer #1: This is a beautiful natural history paper, describing a new reproductive mode for a Neotropical anuran species and other reproductive traits. I really appreciated reading this revised version. The authors did a great job improving the introduction and the discussion. I think that the manuscript can be accepted after minor revision, mostly regarding text corrections and adjustments. Although I tried to correct some grammar errors, I thnk that the manuscript still needs an English revision. My specific suggestions/corrections and comments follow below.

Abstract

Line 47: Include “the” before “novel reproductive mode”.

Introduction

Lines 54-59: In the paper by Silva et al. (2020), authors suggest that the reproductive modes have influenced fecundity and sexual dimorphism in anurans, not the opposite as stated in the introduction. Thus, I suggest rewriting the sentence excluding this part, as follows:

“However, recent studies indicate that sexual selection by male-male competition [5] and parental care [6] also played an important role in the evolution of terrestrial reproductive modes in anuran amphibians, indicating that multiple factors can contribute towards terrestriality.” You can also join the first two paragraphs.

Line 61: replace “tetrapods”

Line 76: replace “…is the diversity in tadpole morphology”

Line 78: substitute “upon” for “on”

Line 85: substitute “to” for “in”

Line 86: delete “categorized as”

Lines 88-90: rewrite as follows: “…and territorial calls may have unique traits and play determinant roles within the social context and reproductive biology of anurans”

Line 95: include “a genus of” before gladiator treefrog

Line 101: include “the” before reproductive biology

Line 107: include “the” before Serra do Mar

Line 116: include “the” before reproduction

Materials and methods

Line 132: it seems that montane is repeated

Line 180: when referring to live animals, it is better to use “individuals, males, females, etc”. Specimen is mainly used for dead animals deposited in collections. Check throughout the text.

Lines 191-192: If you used these tissue samples from adults for tadpole identification, it would be nice to clearly state that “These samples were used for tadpole molecular identification (see details below).”

Line 245: replace “are given”

Line 286: It would be better to write “…traits were measured according to Köhler et al…”

Results

Line 315: exclude “and” before Vriesea and insert a comma

Lines 312-314: You have beautiful pictures, thus you could refer to Fig. 2 when describing calling sites.

Lines 323-326: The first paragraph of the results describes temporal breeding pattern and habitat use. Thus, results on male and female SVL and sexual dimorphism fit better in the second paragraph.

Table 1: Include at the end of the legend: “Days between recaptures are shown as mean (range)”. Exclude “mean (range)” from the table. Also, standardize decimal digits.

Line 349: I suggest “Courtship and oviposition behaviors”

Figure 3: Some adjustments in the legend: “(A) Female approaching the resident male in vocal activity. The white arrow indicates the location of the resident male in the bromeliad leaf-tank. On the left, the satellite male was observing the interaction. (B) In another set of bromeliads, the same female was inspecting the bromeliad leaf-tank while the resident male was in the leaf-tank above (white arrow). (C) Pair in amplexus in the leaf-tank inspected by the female. (D) After egg laying was completed, resident male left the female and sat on the upper leaf-tank, where he began to emit a courtship call to attract the female again. (E) Pair in amplexus in the upper leaf-tank. Notice the spawn just placed in the anterior leaf-tank and the approach of the satellite male (white arrow). (F) Satellite male sitting on the newly deposited spawn (white arrow). The pair remained in amplexus in the leaf-tank above”.

Line 396: substitute “couple” for “pair”. In the same line, substitute “the couple were” for “the pair was”. Check throughout the text.

Line 431: correct “Spawns were found”

Figures 5, 6: It would be nice if photos could be arranged in the same order they are mentioned in the text, or maybe change the order in the text? For instance, at the beginning of the tadpole description (line 483), it is possible to mention Fig 5A-D.

Line 581: There is no Fig 8 anymore. To which figure do you refer here?

Table 3: “Acoustic parameters of the calls of Bokermannohyla astartea…”

Discussion

Lines 688-695: It is always nice to initiate the discussion calling attention for the main results. Thus, I suggest beginning with the discussion about the use of bromeliads as calling and breeding sites and the record of a bromeligenous species in the tribe Cophomantini for the first time. It is easy to be done by changing the position of the sentences.

Lines 685-686: Discussion on the reproductive period should be moved to the end of the paragraph. Moreover, it would be nice to add a sentence about the occurrence of the species in the Atlantic forest, where annual precipitation is high mainly in spring/summer, which may favor the prolonged breeding season of the species and the use of bromeliads to reproduce.

Lines 686-687: As I mentioned in the results, sexual size dimorphism should be discussed in the second paragraph. Males and females exhibited differences in SVL. Which differences? Clearly state that females were larger and mention that this is common pattern for anurans (with some references), although other Bokermannohyla species do not present sexual size dimorphism.

Line 719: substitute “positioning themselves over” by “sitting on”

Lines 720-722: rewrite as follows: “…when opportunistic males try to fertilize oocytes released by the amplectant female, which were not fertilized by the resident male...”

Line 725: I suggest “Courtship and oviposition behaviors”

Line 732: substitute “conduct” by “conducting”

Line 737: include “possibly” before “because”

Line 741-742: Courtship precedes amplexus and oviposition, thus I suggest rewriting as: “We recorded the partitioning of spawns among different leaf-tanks by amplectant females of B. astartea.”

Line 748: Partitioning of spawns has also been recorded for the hylid Dendropsophus haddadi (Silva et al. 2019. Reproductive biology of Dendropsophus haddadi (Bastos and Pombal, 1994), a small treefrog of the Atlantic forest. Herpetology Notes 12: 319-325). Given that Hylidae is a speciose family, I suggest adding that spawn partitioning might be common, although poorly reported.

Line 754: Emphasize your findings about the reproductive mode by adding “novel for anurans” at the end of the sentence.

Line 773: The correct would be Table 2 (tadpole measurements).

Line 774: reword “bromeliad leaf-tanks”.

Lines 777-780: It is intriguing why only tadpoles up to stage 26 occur in bromeliads. Why not until stage 21 or 24? There is a paper by Leite and Eterovick (2010), where they describe the tadpole of Bokermannohyla martinsi and make interesting comparisons among Bokermannohyla tadpoles regarding their morphology and ecology. In this paper, the authors suggest that Bokermannohyla tadpoles have a long duration of stage 25, probably related with the fact that most species reproduce in permanent ponds/streams. Do you think that the long duration of stage 25 could explain why only tadpoles up to stage 26 were found in bromeliads? Or, alternatively, is it possible that tadpoles have a mechanism of stop developing and growing while inside the bromeliads? There are many studies describing plasticity in tadpole growth and development (depending on environmental conditions) and I think it would be interesting to raise some of these hypotheses.

Line 856: substitute “between” for “with”

Line 858: reword: “may also have”

Lines 860-874: As I mentioned in the introduction, there is a misinterpretation of the study by Silva et al. (2020) about reproductive modes and sexual size dimorphism (SSD), and no sexual selection mechanisms are involved. I would suggest removing this discussion on SSD. However, if you decide to keep, I suggest reorganizing this paragraph, as follows:

“… some traits of the reproductive biology of B. astartea agree with a recent complementary hypothesis that is linked to sexual selection [5]. For example, the territoriality of males related to the sets of bromeliads and hidden amplexus and spawning may decrease the risk of polyandry [37, 5] and can also reduce spawning damage by multiple male harassment [5]. Besides male-male competition, terrestrial reproductive modes have been suggested to influence sexual size dimorphism and fecundity in anurans [7]. The less pronounced sexual dimorphism in relation to the SVL between males and females of B. astartea may occur because amplectant females do not carry males to another site for spawning [7]. Still, this space limitation of bromeliad leaf-tanks may favor the reduction of female size since the pair has to fit in the small space to spawn (e.g., [37, 100]), possibly reducing spawning size in these microhabitats, and thus resulting in a less pronounced sexual size dimorphism [7]. These traits are postulated to be strongly related to the evolution and diversification of terrestrial/arboreal reproductive modes in frogs, and reinforces that other selective pressures can act beyond predation and competition [5, 7].”

Line 875: include “anuran” before “species”

Line 879: include “the” before “Atlantic forest”

Line 881: I suggest “…that favor the occurrence of many types of humid microhabitats…”

Line 885: I suggest “…basic biology of Neotropical anurans…”

7. PLOS authors have the option to publish the peer review history of their article (what does this mean?). If published, this will include your full peer review and any attached files.

Reviewer #1: No

---

## [Author Response · Author response to Decision Letter 1]

17 Jan 2021

Editor:

1. Thank you for submitting your manuscript to PLOS ONE. After careful consideration, we feel that it has merit but does not fully meet PLOS ONE’s publication criteria as it currently stands. Therefore, we invite you to submit a revised version of the manuscript that addresses the points raised during the review process.

There are some few issues left that need attention. The one referee and myself were very satisfied with your review.

Answer: Thank you for all suggestions and corrections. We seek to address most of the issues raised, which we present below. We also made a new English revision.

Reviewer #1

1. This is a beautiful natural history paper, describing a new reproductive mode for a Neotropical anuran species and other reproductive traits. I really appreciated reading this revised version. The authors did a great job improving the introduction and the discussion. I think that the manuscript can be accepted after minor revision, mostly regarding text corrections and adjustments. Although I tried to correct some grammar errors, I thnk that the manuscript still needs an English revision. My specific suggestions/corrections and comments follow below.

Answer: We are very grateful for your review that improved the quality of our manuscript. We carefully reviewed each question and suggestions, trying to include them in the revised version of manuscript. For issues that have not been fully addressed, we have justified and discussed them thoroughly. We also made a new English revision.

Abstract

2. Line 47: Include “the” before “novel reproductive mode”.

Answer: Done.

Introduction

3. Lines 54-59: In the paper by Silva et al. (2020), authors suggest that the reproductive modes have influenced fecundity and sexual dimorphism in anurans, not the opposite as stated in the introduction. Thus, I suggest rewriting the sentence excluding this part, as follows:

“However, recent studies indicate that sexual selection by male-male competition [5] and parental care [6] also played an important role in the evolution of terrestrial reproductive modes in anuran amphibians, indicating that multiple factors can contribute towards terrestriality.” You can also join the first two paragraphs.

Answer: Done. As suggested, we rewrote the sentence and join the first two paragraphs.

4. Line 61: replace “tetrapods”

Answer: Done.

5. Line 76: replace “…is the diversity in tadpole morphology”

Answer: Done.

6. Line 78: substitute “upon” for “on”

Answer: Done.

7. Line 85: substitute “to” for “in”

Answer: Done.

8. Line 86: delete “categorized as”

Answer: Done.

9. Lines 88-90: rewrite as follows: “…and territorial calls may have unique traits and play determinant roles within the social context and reproductive biology of anurans”

Answer: Done. We rewrited the sentence as suggested.

10. Line 95: include “a genus of” before gladiator treefrog

Answer: Done.

11. Line 101: include “the” before reproductive biology

Answer: Done.

12. Line 107: include “the” before Serra do Mar

Answer: Done.

13. Line 116: include “the” before reproduction

Answer: Done.

Materials and methods

14. Line 132: it seems that montane is repeated

Answer: Done. We excluded the repeated word.

15. Line 180: when referring to live animals, it is better to use “individuals, males, females, etc”. Specimen is mainly used for dead animals deposited in collections. Check throughout the text.

Answer: Done. We corrected the use of the word “specimen” throughout the text. Then, as suggested, we used “exemplars, individuals, males, etc” to refer to live animals.

16. Lines 191-192: If you used these tissue samples from adults for tadpole identification, it would be nice to clearly state that “These samples were used for tadpole molecular identification (see details below).”

Answer: Done. We included the suggested sentence.

17. Line 245: replace “are given”

Answer: Done.

18. Line 286: It would be better to write “…traits were measured according to Köhler et al…”

Answer: Done.

Results

19. Line 315: exclude “and” before Vriesea and insert a comma

Answer: Done.

20. Lines 312-314: You have beautiful pictures, thus you could refer to Fig. 2 when describing calling sites.

Answer: Done.

21. Lines 323-326: The first paragraph of the results describes temporal breeding pattern and habitat use. Thus, results on male and female SVL and sexual dimorphism fit better in the second paragraph.

Answer: Done. We included this part of the results in a second paragraph.

22. Table 1: Include at the end of the legend: “Days between recaptures are shown as mean (range)”. Exclude “mean (range)” from the table. Also, standardize decimal digits.

Answer: Done.

23. Line 349: I suggest “Courtship and oviposition behaviors”

Answer: Done.

24. Figure 3: Some adjustments in the legend: “(A) Female approaching the resident male in vocal activity. The white arrow indicates the location of the resident male in the bromeliad leaf-tank. On the left, the satellite male was observing the interaction. (B) In another set of bromeliads, the same female was inspecting the bromeliad leaf-tank while the resident male was in the leaf-tank above (white arrow). (C) Pair in amplexus in the leaf-tank inspected by the female. (D) After egg laying was completed, resident male left the female and sat on the upper leaf-tank, where he began to emit a courtship call to attract the female again. (E) Pair in amplexus in the upper leaf-tank. Notice the spawn just placed in the anterior leaf-tank and the approach of the satellite male (white arrow). (F) Satellite male sitting on the newly deposited spawn (white arrow). The pair remained in amplexus in the leaf-tank above”.

Answer: Done.

25. Line 396: substitute “couple” for “pair”. In the same line, substitute “the couple were” for “the pair was”. Check throughout the text.

Answer: Done. We replaced the word "couple" for the word "pair" throughout the text.

26. Line 431: correct “Spawns were found”

Answer: Done.

26. Figures 5, 6: It would be nice if photos could be arranged in the same order they are mentioned in the text, or maybe change the order in the text? For instance, at the beginning of the tadpole description (line 483), it is possible to mention Fig 5A-D.

Answer: Accordingly, we mentioned Fig 5A-D at the beginning of the tadpole description (line 483). Regarding to Fig 6 we tried to arrange the photos in the same order they are called in the text. However, in the case of figures 5 and 6, this is difficult because in the description we always refer to a general aspect of the tadpole (Fig 5), but describing the details of characters (Fig 6) in the sequence. To solve this, we would have to change the description, making it more redundant, or merge the figures 5 and 6, which would be unfeasible due to the number of photos. We think that this problem can be minimized if the plates are positioned near each other in the final version.

27. Line 581: There is no Fig 8 anymore. To which figure do you refer here?

Answer: The Fig 8 has been incorporated into Fig 5, in this case the reference is Fig 5D. We corrected it in the text.

28. Table 3: “Acoustic parameters of the calls of Bokermannohyla astartea…”

Answer: Done.

Discussion

29. Lines 688-695: It is always nice to initiate the discussion calling attention for the main results. Thus, I suggest beginning with the discussion about the use of bromeliads as calling and breeding sites and the record of a bromeligenous species in the tribe Cophomantini for the first time. It is easy to be done by changing the position of the sentences.

Answer: Done. We moved this part of the text to the beginning of discussion.

30. Lines 685-686: Discussion on the reproductive period should be moved to the end of the paragraph. Moreover, it would be nice to add a sentence about the occurrence of the species in the Atlantic forest, where annual precipitation is high mainly in spring/summer, which may favor the prolonged breeding season of the species and the use of bromeliads to reproduce.

Answer: Done. We moved this part of the text to the end of the paragraph and added a sentence about the seasonality of the species as suggested.

31. Lines 686-687: As I mentioned in the results, sexual size dimorphism should be discussed in the second paragraph. Males and females exhibited differences in SVL. Which differences? Clearly state that females were larger and mention that this is common pattern for anurans (with some references), although other Bokermannohyla species do not present sexual size dimorphism.

Answer: Done. We moved this part of the text to the second paragraph, and improved the text with discussion about the differences in SVL between males and females of B. astartea.

32. Line 719: substitute “positioning themselves over” by “sitting on”

Answer: Done.

33. Lines 720-722: rewrite as follows: “…when opportunistic males try to fertilize oocytes released by the amplectant female, which were not fertilized by the resident male...”

Answer: Done. We rewrote the sentence as suggested. 

34. Line 725: I suggest “Courtship and oviposition behaviors”

Answer: Done.

35. Line 732: substitute “conduct” by “conducting”

Answer: Done.

36. Line 737: include “possibly” before “because”

Answer: Done.

37. Line 741-742: Courtship precedes amplexus and oviposition, thus I suggest rewriting as: “We recorded the partitioning of spawns among different leaf-tanks by amplectant females of B. astartea.”

Answer: Done.

38. Line 748: Partitioning of spawns has also been recorded for the hylid Dendropsophus haddadi (Silva et al. 2019. Reproductive biology of Dendropsophus haddadi (Bastos and Pombal, 1994), a small treefrog of the Atlantic forest. Herpetology Notes 12: 319-325). Given that Hylidae is a speciose family, I suggest adding that spawn partitioning might be common, although poorly reported.

Answer: Done. We have included the information and the respective reference in the manuscript.

39. Line 754: Emphasize your findings about the reproductive mode by adding “novel for anurans” at the end of the sentence.

Answer: Done.

40. Line 773: The correct would be Table 2 (tadpole measurements).

Answer: Done. We corrected it in the text.

41. Line 774: reword “bromeliad leaf-tanks”.

Answer: Done. We corrected it in the text.

42. Lines 777-780: It is intriguing why only tadpoles up to stage 26 occur in bromeliads. Why not until stage 21 or 24? There is a paper by Leite and Eterovick (2010), where they describe the tadpole of Bokermannohyla martinsi and make interesting comparisons among Bokermannohyla tadpoles regarding their morphology and ecology. In this paper, the authors suggest that Bokermannohyla tadpoles have a long duration of stage 25, probably related with the fact that most species reproduce in permanent ponds/streams. Do you think that the long duration of stage 25 could explain why only tadpoles up to stage 26 were found in bromeliads? Or, alternatively, is it possible that tadpoles have a mechanism of stop developing and growing while inside the bromeliads? There are many studies describing plasticity in tadpole growth and development (depending on environmental conditions) and I think it would be interesting to raise some of these hypotheses.

Answer: This is an interesting topic. They probably occur in bromeliads up to stage 26, because after that they would reach a maximum size unfeasible for living in small space. We added in the results information about the development stage of the tadpoles when they hatch of the eggs in bromeliads (stages 23-24). Thus, our data shows that only the initial and small free-living stages 23-26 would occur in bromeliads. The development/growth patterns mentioned for B. martinsi also occur in some species of the B. pseudopseudis group (Cardoso 1983; Eterovick & Brandão 2001; Lugli & Haddad 2006b; Lins et al 2018) but were not observed in species of the B. circumdata group (e.g., Mogin & Carvalho-e-Silva, 2013, Pezzuti et al. 2015). In species of this group, the correlation between development and growth seems to be more concerted. As we did not present data of development and growth, nor observed these patterns in nature, we are unaware to discuss which patterns B. astartea presents. Anyway, as we did not find large tadpoles in stage 25, it is unfeasible that they retain this stage for a long time. Although it is an interesting topic, we think that discussing this topic (on development and growth) without presenting these data, would increase the size of the manuscript, without bringing new information. 

43. Line 856: substitute “between” for “with”

Answer: Done.

44.Line 858: reword: “may also have”

Answer: Done.

45. Lines 860-874: As I mentioned in the introduction, there is a misinterpretation of the study by Silva et al. (2020) about reproductive modes and sexual size dimorphism (SSD), and no sexual selection mechanisms are involved. I would suggest removing this discussion on SSD. However, if you decide to keep, I suggest reorganizing this paragraph, as follows:

“… some traits of the reproductive biology of B. astartea agree with a recent complementary hypothesis that is linked to sexual selection [5]. For example, the territoriality of males related to the sets of bromeliads and hidden amplexus and spawning may decrease the risk of polyandry [37, 5] and can also reduce spawning damage by multiple male harassment [5]. Besides male-male competition, terrestrial reproductive modes have been suggested to influence sexual size dimorphism and fecundity in anurans [7]. The less pronounced sexual dimorphism in relation to the SVL between males and females of B. astartea may occur because amplectant females do not carry males to another site for spawning [7]. Still, this space limitation of bromeliad leaf-tanks may favor the reduction of female size since the pair has to fit in the small space to spawn (e.g., [37, 100]), possibly reducing spawning size in these microhabitats, and thus resulting in a less pronounced sexual size dimorphism [7]. These traits are postulated to be strongly related to the evolution and diversification of terrestrial/arboreal reproductive modes in frogs, and reinforces that other selective pressures can act beyond predation and competition [5, 7].”

Answer: Thank you for this correction in the text. We corrected the part that had been misinterpreted and decided to keep the mention of SSD in the discussion, considering your suggested text.

46. Line 875: include “anuran” before “species”

Answer: Done.

47. Line 879: include “the” before “Atlantic forest”

Answer: Done.

48. Line 881: I suggest “…that favor the occurrence of many types of humid microhabitats…”

Answer: Done.

49. Line 885: I suggest “…basic biology of Neotropical anurans…”

Answer: Done.

---

## [Editor Report · Decision Letter 2]

19 Jan 2021

A new reproductive mode in anurans: natural history of Bokermannohyla astartea (Anura: Hylidae) with the description of its tadpole and vocal repertoire

PONE-D-20-19802R2

Dear Dr. Malagoli,

We’re pleased to inform you that your manuscript has been judged scientifically suitable for publication and will be formally accepted for publication once it meets all outstanding technical requirements.

Kind regards,

Stefan Lötters

Academic Editor

PLOS ONE
---

## [Editor Report · Acceptance letter]

22 Jan 2021

PONE-D-20-19802R2 

A new reproductive mode in anurans: natural history of *Bokermannohyla astartea* (Anura: Hylidae) with the description of its tadpole and vocal repertoire 

Dear Dr. Malagoli:

I'm pleased to inform you that your manuscript has been deemed suitable for publication in PLOS ONE. Congratulations! Your manuscript is now with our production department. 

Kind regards, 

on behalf of

Prof. Dr. Stefan Lötters 

Academic Editor

PLOS ONE